# Energy deficiency selects crowded live epithelial cells for extrusion

Saranne J. Mitchell[1,2,3,4], Carlos Pardo-Pastor[1,2,4,5], Anastassia Tchoumakova[1,2,4], Thomas A. Zangle[6] & Jody Rosenblatt[1,2,4 ✉]

Epithelial cells work collectively to provide a protective barrier, yet they turn over rapidly through cell division and death. If the numbers of dividing and dying cells do not match, the barrier can vanish, or tumours can form. Mechanical forces through the stretch-activated ion channel Piezo1 link both of the processes; stretch promotes cell division, whereas crowding triggers live cells to extrude and then die[1,2]. However, it was not clear what selects a given crowded cell for extrusion. Here we show that the crowded cells with the least energy and membrane potential are selected for extrusion. Crowding triggers sodium ($Na^+$) entry through the epithelial $Na^+$ channel (ENaC), which depolarizes cells. While those with sufficient energy repolarize, those with limited ATP remain depolarized, which, in turn, triggers water egress through the voltage-gated potassium ($K^+$) channels $K_v1.1$ and $K_v1.2$ and the chloride ($Cl^-$) channel SWELL1. Transient water loss causes cell shrinkage, amplifying crowding to activate crowding-induced live cell extrusion. Thus, our findings suggest that ENaC acts as a tension sensor that probes for cells with the least energy to extrude and die, possibly damping inadvertent crowding activation of Piezo1 in background cells. We reveal crowding-sensing mechanisms upstream of Piezo1 that highlight water regulation and ion channels as key regulators of epithelial cell turnover.

Cell extrusion is a conserved mechanism for maintaining cell number homeostasis, driving cell death in epithelia from sea sponges to humans. During extrusion, a live cell is ejected apically by coordinated basolateral actomyosin contraction of the extruding cell and its neighbours[2]. Once extruded, a cell will die from lack of survival signalling. We previously found that Piezo1 activates live-cell extrusion (LCE) in response to crowding to maintain constant epithelial cell numbers[1,2]. Crowding activates Piezo1 to trigger a canonical pathway that relies on secretion of the lipid sphingosine 1-phosphate (S1P), which binds to the G-protein-coupled receptor $S1P_2$ to activate Rho-mediated actomyosin contraction needed for extrusion[1,3,4]. This same $S1P–S1P_2–Rho$ pathway also extrudes apoptotic cells to ensure that no gaps are formed in the monolayer[5]. As crowding-induced cell extrusion drives most epithelial cell death, identifying which cells within a crowded epithelial field extrude is central to understanding what governs cell death. In sparse epithelia, topological defects perturbing normal epithelial hexagonal packing promote outlier cells to extrude[6,7]. Moreover, epithelial cells with replicative stress in *Caenorhabditis elegans* and mammals are also targeted for extrusion[7,8]. However, as most cell extrusions occur in crowded regions, where cells are not dividing and have no obvious topological defects, what marks most cells for extrusion is still unclear.

As mechanical cell competition suggested that differences in strain resistance might select cells for extrusion[9], we first considered whether cells with less mass are selected under compressive forces, as they may be less resistant to strain. To test this possibility, we adapted quantitative phase imaging (QPI) to analyse dry-mass changes over time in Madin–Darby canine kidney II (MDCKII) epithelial monolayers[10,11]. However, we found that dry mass does not change before cell extrusion. By contrast, dry mass increases before a cell divides (Extended Data Fig. 1a–c and Supplementary Video 1; representing 31 cells from 12 total cell islands), disfavouring reduced cell mass as a selecting factor for LCE.

Another possibility is that cells with the highest compression might activate a calcium ($Ca^{2+}$) wave through Piezo1, the most upstream known regulator of extrusion. We previously identified a single Piezo1-dependent $Ca^{2+}$ spark occurred in epithelial cells before their division[1]. To test whether $Ca^{2+}$ spikes before LCE, we imaged MDCK monolayers transfected with a genetically encoded GFP $Ca^{2+}$ indicator, GECO1, and monitored extrusion using phase microscopy. We found no discernible $Ca^{2+}$ spikes or waves before LCE and, instead, we noticed a dip before extrusion (Extended Data Fig. 1d,e). While the 1 s imaging interval might have missed shorter $Ca^{2+}$ transients, our results suggest that $Ca^{2+}$ is not linked to extrusion.

## Volume loss leads to LCE

However, during these experiments, we noticed a consistent transient increase in cell–cell junction brightness lasting about 6.5 min before a cell extrudes (Fig. 1a,b and Supplementary Video 2). Increased phase brightness might indicate a decrease in cell volume or cell

[1]The Randall Centre for Cell & Molecular Biophysics, School of Basic & Medical Biosciences, King's College London, London, UK. [2]School of Cancer and Pharmaceutical Sciences, King's College London, London, UK. [3]Department of Biomedical Engineering, University of Utah, Salt Lake City, UT, USA. [4]Francis Crick Institute, London, UK. [5]Laboratory of Molecular Physiology, Department of Medicine and Life Sciences, Universitat Pompeu Fabra, Barcelona, Spain. [6]Department of Chemical Engineering, University of Utah, Salt Lake City, UT, USA. ✉e-mail: jody.rosenblatt@kcl.ac.uk

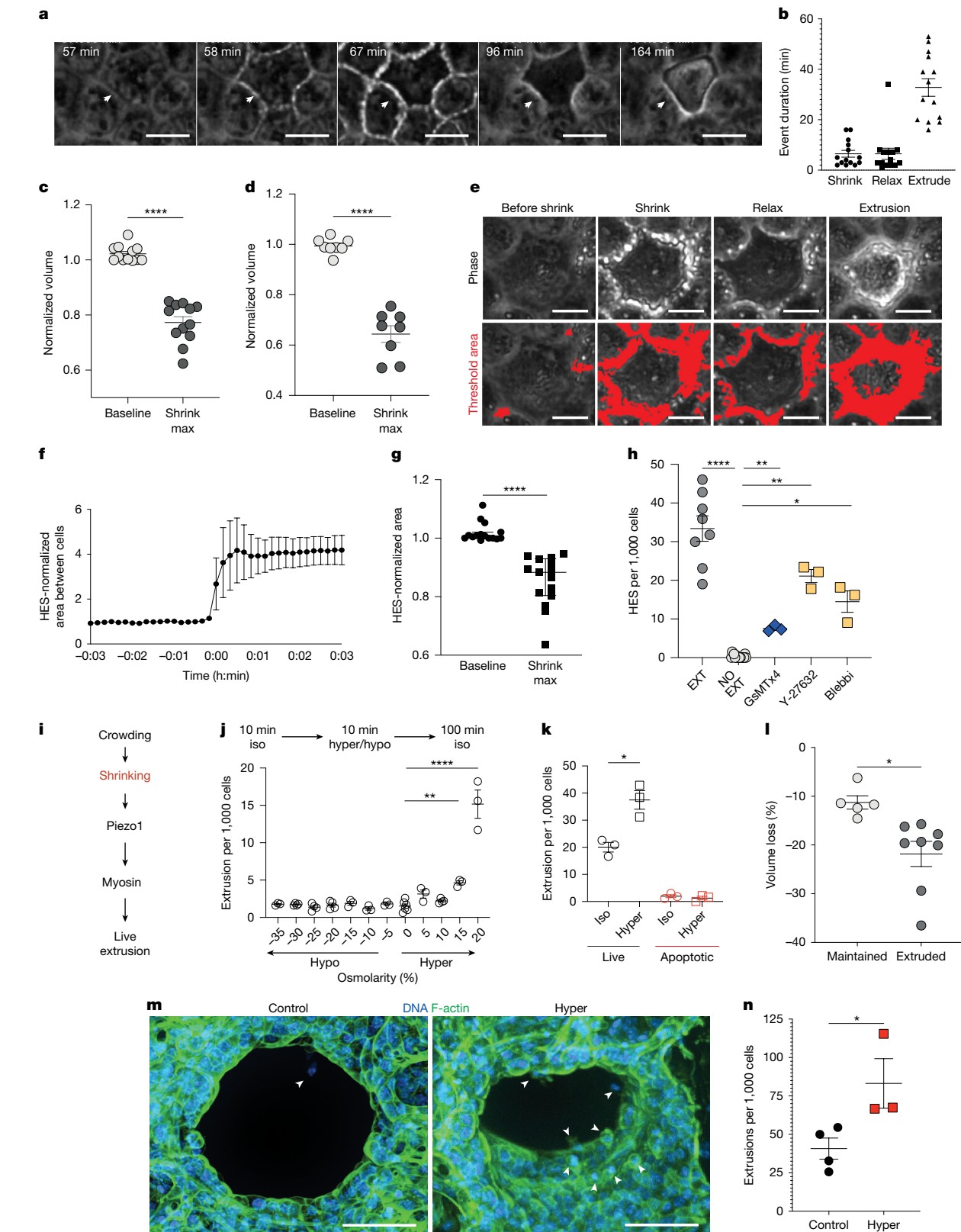

**Fig. 1** | See next page for caption.

shape change. To test whether transient volume loss accounts for the phase brightness before extrusion, we mosaically expressed cytoplasmic GFP in MDCKII cells and measured the cell volume using confocal time-lapse microscopy before and during extrusion. This approach indicated that live cells experience a transient volume

loss before extruding that directly corresponded to the junctional brightness measured in our phase-contrast time lapses (Fig. 1a–c). To confirm the volume reduction using an alternative method, we used Calcein-AM, which fluoresces green but is quenched as cells lose water and shrink[12]. Calcein-AM confirmed that cells lose volume

before extruding (Fig. 1d), which we refer to as homeostatic early shrinkage (HES).

**Fig. 1 | Live cells shrink briefly through water loss before extruding in dense regions. a**, Phase time-lapse stills show junctional lightning (arrowhead) around a live cell before extrusion. Scale bars, 10 μm. $n = 5$. **b**, The mean ± s.e.m. duration that cells experience shrinking, relaxation and onset of shrinking to extrusion. $n = 16$ cells. **c**, Shrinking based on GFP. The mean ± s.e.m. normalized cell volume at the baseline and after shrinking is shown. $n = 12$ cells. Statistical analysis was performed using unpaired *t*-tests. **d**, Shrinking based on Calcein. The mean ± s.e.m. normalized cell volume at the baseline and after shrinking is shown. $n = 8$ cells. Statistical analysis was performed using unpaired *t*-tests. **e,f**, Representative time-lapse stills from the lightning assay showing the area around cells (red, bottom) before shrinking, after shrinking, at relaxation and at onset of extrusion (**e**) and the mean ± s.e.m. normalized area around cells (**f**), where $t_0$ is the shrinkage. $n = 10$ cells. Scale bars, 5 μm. **g**, The mean ± s.e.m. normalized cell area at the baseline and after shrinking, as determined using a lightning assay. $n = 15$ cells. Statistical analysis was performed using two-way analysis of variance (ANOVA) with Dunnett's multiple-comparison test. **h**, The mean ± s.e.m. rate of cell shrinkage (HES) of cells before extrusion (EXT), of cells remaining in the monolayer (no EXT) or of cells after treatment with extrusion inhibitors (GsMTx4, Y-27632 or blebbistatin (blebbi)), as determined using a lightning assay. Statistical analysis was performed using by one-way ANOVA with Welch multiple-comparison correction; \*\**P* = 0.0026, \*\**P* = 0.0064, \**P* = 0.036; respectively. **i**, Schematic placing shrinkage upstream of the established extrusion pathway. **j**, The mean ± s.e.m. extrusion rate in MDCKII cells incubated under increasing osmolarities. Statistical analysis was performed using one-way ANOVA with Dunnett's multiple-comparison correction; \*\**P* = 0.0068. Hyper, hyperosmolarity; hypo, hypoosmolarity; iso, isotonic. **k**, The mean ± s.e.m. live and apoptotic extrusion rates (using cleaved-caspase indicator) after no (iso) or 20% hypertonic shock of MDCKII cells. Statistical analysis was performed using unpaired *t*-tests; \**P* = 0.0105. **l**, The mean ± s.e.m. cell volume (based on GFP) after 20% hypertonic shock of cells that are maintained or extruded. $n = 5$. Statistical analysis was performed using unpaired *t*-tests; \**P* = 0.0113. **m,n**, Representative mouse precision-cut ex vivo lung slice (PCLS) images of extrusions (**m**; white arrow heads) and the corresponding rates (**n**). Statistical analysis was performed using unpaired *t*-tests; \**P* = 0.0431. Scale bar, 50 μm. All $n = 3$ and independent two-tailed experiments unless otherwise stated. \*\*\*\**P* < 0.0001.

Given that the transient phase brightness increase reflects cell volume loss, we developed the lightning assay—a semi-automated method to readily assay HES, based on thresholding junctional brightness intensity around cells before they extrude (Fig. 1e–g). Using the lightning assay, we found that around 70% of cells shrink before LCE, whereas only <0.03% of cells filmed (8 out of 750) shrink without extruding. Moreover, these few cells that do not extrude were in uncrowded epithelial regions, whereas HES-induced LCE occurred in crowded areas.

As about 70% of extruding cells undergo HES, it was not clear why the remaining 30% do not shrink. One possibility is that cells that do not shrink undergo apoptotic extrusion, which accounts for around 20–30% of extruding cells at steady state[1]. Combining the lightning assay with a fluorescent reporter of the apoptotic marker cleaved caspase-3, we found that cell shrinkage is rare (~3%) before apoptotic extrusion and could represent the 30% of cells that do not shrink (Extended Data Fig. 2a,b and Supplementary Video 3). As most cells undergo HES, which precedes LCE, this became of the focus of our investigation.

To identify what causes HES, we investigated whether known extrusion signals, such as actomyosin contractility and Piezo1 activation, might also affect volume. Notably, contractility, which is required for volume loss during apoptotic extrusion[13,14], is dispensable for HES, as contractility inhibition with Y-27632 or blebbistatin instead increased the number of cells shrinking by about 23× compared with the untreated controls (Fig. 1h). Thus, contractility appears to suppress cell shrinkage, presumably by stabilizing cell–cell junctions. As HES occurs in crowded regions, we tested whether transiently inhibiting stretch-activated ion channels (SACs) such as Piezo1, which is required for crowding-induced LCE[1], prevented HES. Notably, we found that HES still occurs in the presence of the SAC blocker GsMTx4, suggesting that Piezo1—the earliest known mechanosensor to trigger extrusion in response to crowding[1]—acts downstream of cell shrinkage (Fig. 1h). Thus, crowding followed by single-cell shrinkage appear to initiate LCE upstream of Piezo1 and myosin contraction (Fig. 1i).

We next investigated whether water efflux might regulate the transient volume loss cells experience before extrusion. To first test whether water loss is sufficient to initiate LCE, we incubated MDCKII monolayers in medium with altered osmolarity (hypotonic to hypertonic) for 10 min before returning to isotonic medium to mimic the shrink duration seen before extrusion (Supplementary Videos 4 and 5). While hypotonic medium had no effect on cell extrusion (Fig. 1j), 15–20% hypertonicity substantially increased both cell shrinkage and LCE rates within 15 min (Fig. 1j and Supplementary Videos 4 and 5), with higher tonicities destroying the monolayers due to excessive extrusion (data

not shown). Although hypertonicity can provoke apoptotic volume decrease[14,15], the active caspase-3 apoptotic reporter indicated that hypertonic treatment induces only live, and not apoptotic, extrusion (Fig. 1k). Furthermore, the pan-caspase inhibitor zVAD-fmk did not prevent hypertonic-induced extrusions (Extended Data Fig. 2b,c), ruling out apoptotic volume decrease in this process. Notably, although the entire monolayer was treated with 20% hypertonic solution, not all cells shrunk equally. Only cells that shrank 20 ± 3%, as measured by cytoplasmic GFP volume, extrude; those shrinking 11 ± 2.5% do not extrude (Fig. 1l and Extended Data Fig. 2d,e).

To test whether volume loss drives extrusion in other epithelial cells, we used ex vivo mouse lung slices, where bronchial epithelia extrude rapidly following crowding[16]. We found that transient hypertonic treatment increased bronchial epithelial cell extrusion by around twofold within 30 min (Fig. 1m,n). Thus, experimentally reducing cell volume, which we term osmotic-induced cell extrusion (OICE), about 20% is sufficient to trigger individual cell extrusion, potentially mimicking HES.

## Voltage-gated ion channels regulate HES

We first assessed whether any ion channel inhibitors block homeostatic extrusion using a library of K⁺ and Cl⁻ channel inhibitors, scoring extrusions by phase microscopy and immunostaining. Notably, all of the inhibitors tested significantly reduced extrusion rates during normal homeostatic turnover to levels seen with SAC channel inhibition (Fig. 2a). Moreover, the same channel inhibitors blocked OICE (Extended Data Fig. 3a). Although dimethyl sulfoxide, the inhibitor solvent, can alter cell permeability to water[17], it did not significantly affect extrusion rates on its own (Fig. 2a,b). As Cl⁻ channel SWELL1 inhibition causes cell death[18], we added the caspase inhibitor zVAD-fmk to the SWELL1 inhibitor DCPIB in all assays (Fig. 2d and Extended Data Fig. 2c). Notably, 4-AP, a voltage-gated $K_v1.1$ and $K_v1.2$ channel inhibitor, also blocks apoptotic extrusion[19], suggesting a conserved activator for all cell extrusions (Fig. 2a and Extended Data Fig. 3a). It was surprising that so many inhibitors block extrusion, warranting further studies to identify how these channels govern extrusion. However, here we focus solely on which channels might cause the shrinkage step preceding most cell extrusions.

To determine whether K⁺ and Cl⁻ channels control HES, we first tested which inhibitors blocked OICE extrusions. Channels were blocked by inhibitors only during the 10-min hypertonic treatment to help focus on potential targets for those involved in shrinkage (Extended Data Fig. 3b,c). Ruling out SACs, and having identified only $K_v$ and Cl⁻ channels as required during early stages of OICE (Fig. 2c), we next tested which of these channels also regulate HES using our lightning assay as a proxy and calcein to measure cell shrinkage in response

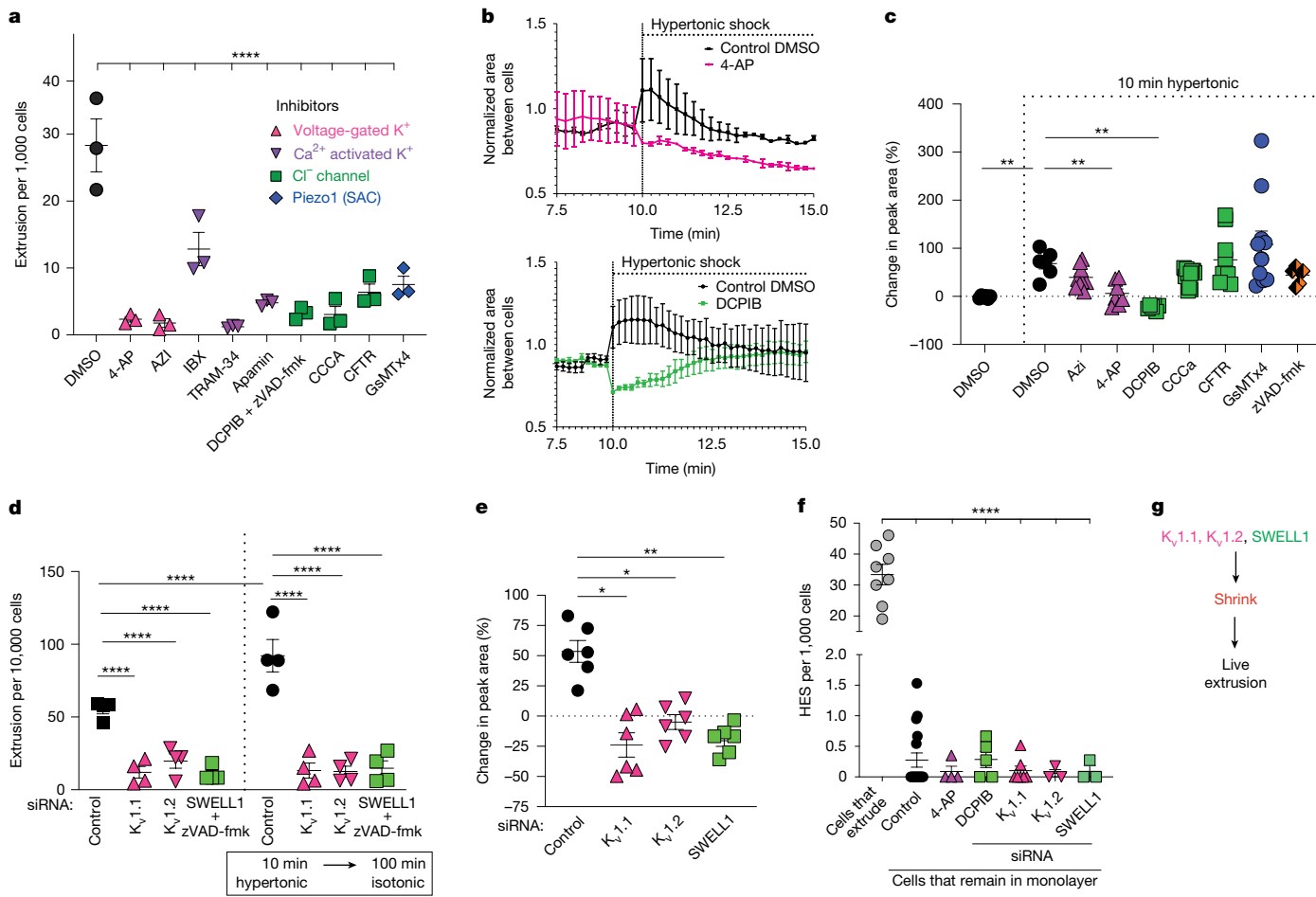

**Fig. 2 | Volume loss is regulated by $K_v1.1$, $K_v1.2$ or SWELL1. a**, The mean ± s.e.m. extrusion rate after treatment with volume-regulating inhibitors, scored from immunostained monolayers. $n = 3$. Statistical analysis was performed using one-way ANOVA with Dunnett's multiple-comparison test. The inhibitor key shows the volume-regulating channel family inhibitor target by assigned colour and icon. **b**, The mean ± s.e.m. area from the lightning assay over time, where increased space around the cell indicates cell shrinkage after treatment with $K_v1.1$ and $K_v1.2$ (4-AP, pink) and SWELL1 (DCPIB, green) inhibitors. $n = 3$. **c**, The mean ± s.e.m. peak area change (from the lightning assay) after treatment with inhibitors during 20% hypertonic challenge. $n = 6$. Statistical analysis was performed using one-way ANOVA with Dunnett's multiple-comparison test; \*\**P* = 0.0020 (DMSO), \*\**P* = 0.0015 (4-AP), \*\**P* = 0.0021 (DCPIB). **d**, The mean ± s.e.m. extrusion rates after siRNA knockdown with or without 20% hypertonic challenge compared with the controls. $n = 4$. All knockdowns blocked extrusion. Statistical analysis was performed using two-way ANOVA with Dunnett's multiple-comparison test. **e**, The mean ± s.e.m. peak area change (as determined using the lightning assay) during 10 min hypertonic challenge of siRNA knockdown cells. $n = 6$. Statistical analysis was performed using one-way ANOVA with Dunnett's multiple-comparison test; \**P* = 0.0162 ($K_v1.1$), \*\**P* = 0.0100 ($K_v1.2$), \*\**P* = 0.0011 (SWELL1). **f**, The mean ± s.e.m. cell shrinkage (HES) rate after treatment with 4-AP or DCPIB, or after $K_v1.1$, $K_v1.2$ or SWELL1 siRNA knockdown, compared with the control background cells or those before extrusion. $n = 3$. Statistical analysis was performed using two-way ANOVA with Dunnett's multiple-comparison test. All $n$ values represent independent two-tailed experiments.

to 20% hypertonicity with or without 1% of channel inhibitors. This assay revealed that only DCPIB (with zVAD-fmk) and 4-AP, which target the $Cl^-$ channel SWELL1 and the voltage-gated $K^+$ channels $K_v1.1$ and $K_v1.2$, respectively, significantly blocked hypertonic-induced cell shrinkage (Fig. 2b,c). By contrast, inhibitors blocking known extrusion signals, Piezo1, S1P and $S1P_2$, do not block cell shrinkage in response to hypertonic solution, similar to the homeostatic outcomes (Fig. 1h and Extended Data Fig. 3d,e). To more specifically test the roles of $K_v1.1$, $K_v1.2$ and SWELL1 in both OICE and HES, we used small interfering RNAs (siRNAs) to knockdown each channel (Extended Data Fig. 3g). Knockdown of any of these channels prevented HES and OICE (Fig. 2d–f). Immunostaining showed that all three channels localize to the cell apex, with $K_v1.1$ co-localizing with ZO-1 at tight and tricellular junctions—regions that are important to tension sensing in epithelia[20] (Extended Data Fig. 3f). Thus, $K_v1.1$, $K_v1.2$ and SWELL1 regulate both cell shrinkage and extrusion during normal cell turnover (Fig. 2g).

Given that $K_v1.1$ and $K_v1.2$ channels initiating cell shrinkage and extrusion are voltage gated, we considered whether membrane depolarization may initiate epithelial cell extrusion. The $Na^+/K^+$ ATPase is essential to generate and maintain the $Na^+$ and $K^+$ electrochemical gradients across the membrane that give rise to resting membrane potential[21] and therefore sets a voltage that can be depolarized. While neuronal potential has primarily been studied, all cells spend significant amounts of total ATP using the $Na^+/K^+$ ATPase to generate and maintain a resting cell membrane potential[21]. Neurons maintain a membrane potential of around −70 mV, while epithelia range between −30 and −50 mV (refs. 22,23). To study whether depolarization precedes shrinkage, we filmed MDCKII monolayers loaded with the fluorescent dye $DiBAC_4(3)$, which becomes brighter with plasma membrane depolarization[24,25]. We found that cells depolarize on average around 5 min before shrinking and extruding (Fig. 3a–c and Extended Data Fig. 4b). Moreover, non-shrinking cells showed no depolarization signs, linking shrinkage and depolarization (Fig. 3a,b and Supplementary Video 6). To test

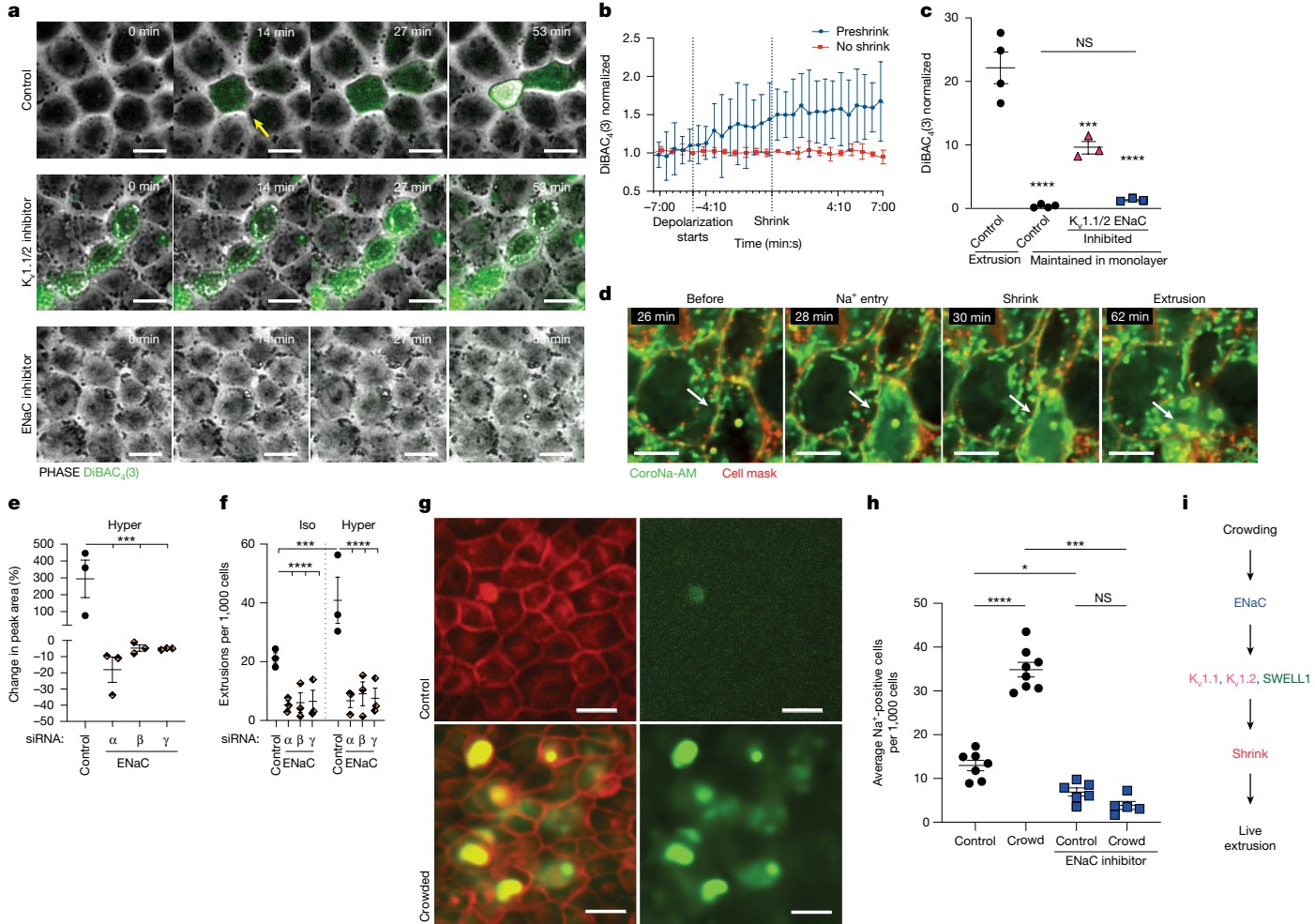

**Fig. 3 | ENaC senses crowding, triggering voltage to induce extrusion.**
**a**, Time-lapse (min) analysis of membrane depolarization before homeostatic extrusion (top; the yellow arrow indicates a cell before extrusion) in the presence of $K_v1.1$ and $K_v1.2$ inhibitor (4-AP) or with ENaC inhibitor (amiloride). Scale bars, 10 μm. $n = 6$. **b**, The mean ± s.e.m. normalized $DiBAC_4(3)$ fluorescence of cells experiencing shrinking before extrusion (blue) or no shrinking (red) over time. $n = 6$. **c**, The mean ± s.e.m. $DiBAC_4(3)$ fluorescence of cells before extrusion under steady-state conditions compared with non-extruding cells or those treated with $K_v1.1$ and $K_v1.2$ (4-AP) or ENaC (amiloride) inhibitors. Statistical analysis was performed using one-way ANOVA with Tukey's multiple-comparison test; ***$P = 0.0008$; NS, not significant. **d**, Representative images over time showing $Na^+$ entry into a cell (white arrow) before shrinking and extruding. Scale bars, 10 μm. $n = 4$. **e**, The mean ± s.e.m. percentage area increase between cells (by lightning assay) after siRNA-mediated ENaC-α, ENaC-β or ENaC-γ knockdown with or without 20% hypertonic challenge; non-targeted siRNAs were used as controls. Statistical analysis was performed using two-way ANOVA with Dunnett's multiple-comparison test; ***$P = 0.0004$ (ENaC-α), ***$P = 0.0006$ (ENaC-β and ENaC-γ). **f**, The mean ± s.e.m. extrusion rates of cells after ENaC-α, ENaC-β or ENaC-γ knockdown with or without 20% hypertonic challenge; non-targeted siRNAs were used as controls. Statistical analysis was performed using two-way ANOVA with Dunnett's multiple-comparison test; ***$P = 0.0004$. **g**, Representative images of cellular $Na^+$ entry with or without crowding. Scale bars, 20 μm. **h**, The mean ± s.e.m. $Na^+$ entry per cell with or without crowding and/or ENaC inhibition with amiloride. Statistical analysis was performed using one-way ANOVA with Šidák's multiple-comparison test; *$P = 0.0131$, ***$P = 0.0005$. **i**, Schematic showing that crowding activates ENaC-dependent $Na^+$ entry upstream of $K_v1.1/K_v1.2$- and SWELL1-dependent cell shrinkage. $n = 3$ and independent two-tailed experiments for all unless otherwise indicated.

whether $K^+$ channel activation is sufficient to trigger cell shrinkage and extrusion, we optogenetically activated $K^+$ channels in specific cells within MDCKII monolayers expressing Blink2 (ref. 26). However, we found that only 1 cell out of 29 activated with blue light went on to shrink and extrude. Thus, activating $K^+$ egress was not sufficient to induce extrusion and suggested that another signal must also operate.

## Mechanical crowding activates ENaC

Typically, $Na^+$ channel activation causes membrane depolarization[21,27,28]; thus, we next investigated whether $Na^+$ ingress activates membrane depolarization and voltage-gated $K^+$ channels. To test whether $Na^+$ entry precedes cell shrinkage and extrusion, we filmed cell monolayers loaded with the fluorescent $Na^+$ dye CoroNa-AM and found that

$Na^+$ entry precedes both HES and extrusion (Fig. 3d). One attractive candidate to mediate $Na^+$ entry upstream of $K_v1.1$ and $K_v1.2$ activation is ENaC—a highly conserved, mechanically activated, apically localized (Extended Data Fig. 4a) channel that causes depolarization through $Na^+$ entry[29,30]. Indeed, inhibiting ENaC with amiloride prevented depolarization, whereas inhibiting $K_v1.1$ and $K_v1.2$ with 4-AP did not (Fig. 3a,c and Extended Data Fig. 4b). Moreover, inhibiting ENaC with amiloride or knocking down any of its subunits (α, β or γ) prevented both hypertonic cell shrinkage and both homeostatic and hypertonic extrusions (Fig. 3e,f and Extended Data Fig. 4c–f). While ENaC is a mechanosensitive $Na^+$ channel, most studies have examined its response to stretch[29]; it is still unknown whether it can respond to the crowding forces that trigger extrusion. Growing cells to homeostatic densities on a stretched PDMS substrate and releasing them from stretch causes

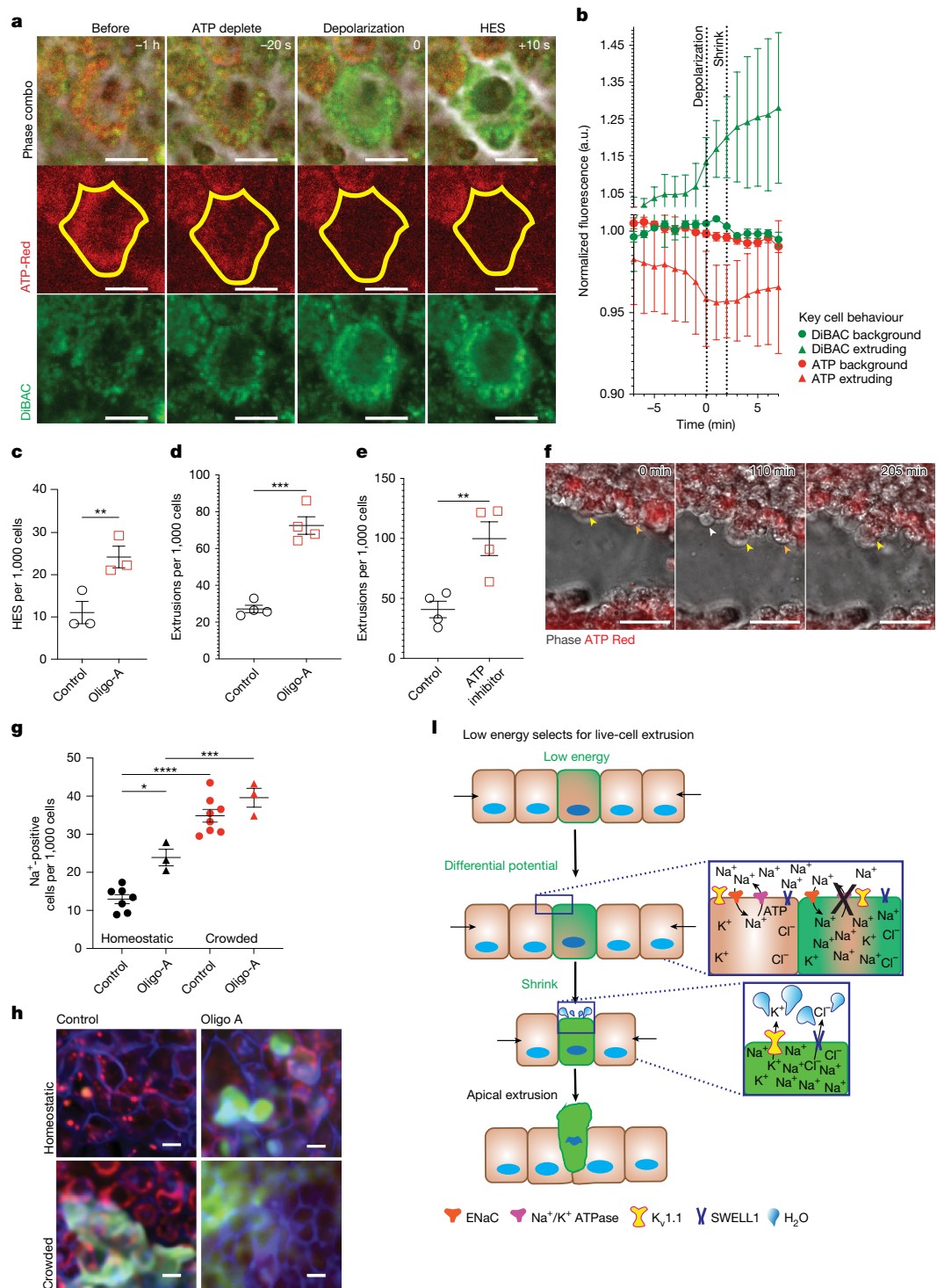

**Fig. 4 | Low energy selects for LCE. a**, Representative phase, membrane depolarization and cellular ATP levels during shrinkage and extrusion of MDCKII cells. Scale bar, 10 μm. **b**, DiBAC$_4$(3) and ATP-Red fluorescence in extruding and non-extruding MDCKII cells. The dotted lines are reference times of depolarization and shrinkage. $n = 3$ (10 cells). **c,d**, The mean ± s.e.m. rates of MDCKII cells exhibiting shrinkage (**c**) and extrusion (**d**) after ATP reduction with oligomycin-A. $n = 3$ (**c**) and $n = 4$ (**d**). Statistical analysis was performed using two-way ANOVA with Dunnett's multiple-comparisons test (**P = 0.0019) or unpaired $t$-tests (***P = 0.0001). **e**, The mean ± s.e.m. epithelial extrusion rates in mouse lung slices with ATP inhibition. Oligo-A, oligomycin-A. $n = 4$. Statistical analysis was performed using unpaired $t$-tests; **P = 0.0093. **f**, Representative time-lapse images of mouse lung epithelial cells stained with ATP-Red undergoing extrusion (white, yellow and orange arrowheads). Scale bar,

25 μm. **g**, The mean ± s.e.m. normalized Na$^+$ entry based on CoroNa-AM fluorescence after ATP inhibition under homeostatic conditions versus during crowding before extrusion of MDCKII cells. $n = 3$. Statistical analysis was performed using one-way ANOVA with Šidák's multiple-comparison test; **P = 0.0047 (homeostatic, oligomycin-A), ***P = 0.0008. **h**, Representative images of Na$^+$ (CoroNa-AM) entry in MDCKII cells under homeostatic and crowding conditions with or without ATP inhibition. Scale bar, 10 μm. **i**, The extrusion pathway model, showing that entry of Na$^+$ ions through ENaC occurs in crowded cells, and this cannot be rectified by the Na$^+$/K$^+$ ATPase pump in cells with low energy. The resulting depolarization activates K$_v$1.1 and K$_v$1.2 (and potentially SWELL1) to trigger a threshold of cell shrinkage that then activates extrusion. All $n$ values represent independent two-tailed experiments.

rapid crowding, we found that this acute crowding increases Na$^+$ entry in an ENaC-dependent manner (Fig. 3g,h). Together, these results suggest that crowding activates ENaC-dependent Na$^+$ entry, which drives membrane depolarization, therefore activating the voltage-gated K$^+$ channels K$_v$1.1 and K$_v$1.2 that are responsible for cell shrinkage (Fig. 3i).

## Low ATP selects for LCE

When following cellular Na$^+$ entry after experimental crowding, we noted that CoroNa-AM fluorescence increased slightly in all cells, but it remained high in cells that later shrink and extruded. Thus, identifying why this subset retains Na$^+$ could account for how crowded cells are selected to extrude. Given that the Na$^+$/K$^+$ ATPase, which is essential to repolarizing the plasma membrane, accounts for 30–70% of all cellular ATP consumption, depending on the cell type[21,31,32], we next investigated whether cells that cannot restore Na$^+$ levels have low ATP. Time-lapse microscopy analysis of cells loaded with ATP-Red (a dye that stains for mitochondrial ATP) and DiBAC$_4$(3) revealed that ATP levels decrease before cells depolarize, shrink and extrude (Fig. 4a,b). A reduction in ATP was confirmed with the genetically encoded ratiometric ATP sensor Queen37 (refs. 33,34) (Extended Data Fig. 5a,b). Furthermore, reducing ATP with the mitochondrial ATP synthase inhibitor oligomycin A increased the number of cells both shrinking and extruding compared with the controls (Fig. 4c,d). Moreover, we tried to reduce Na$^+$/K$^+$ using the inhibitor oubain; however, this killed all epithelial cells rapidly (data not shown). To confirm that low energy driving extrusion was not limited to cultured MDCK monolayers, we filmed bronchial epithelia in mouse ex vivo lung slices, with and without oxamate and oligomycin A to decrease ATP. Again, we found that ATP decreases in cells before they extrude during homeostasis and that experimentally reducing ATP more than doubled extrusion rates (Fig. 4e,f). Inhibition of Piezo1 with GsMTx4 did not block ATP reduction and membrane depolarization, indicating that Piezo1 is downstream in the extrusion pathway (Extended Data Fig. 5c,d). Conversely, supplementing cells with additional glucose decreased the rate of extrusion (Extended Data Fig. 5e). Although depolarization and Na$^+$ entry could not be filmed together, it was found that Na$^+$ entry occurs 2.4 min earlier than depolarization (Extended Data Fig. 5f), further supporting that the Na$^+$ increase does lead to depolarization. Notably, ATP reduction on its own increased intracellular Na$^+$, and this was further increased by crowding (Fig. 4g,h), suggesting that crowding increases Na$^+$ that cannot be pumped back out of cells with lower energy.

Together, our findings suggest a model in which crowding selects cells with insufficient energy for extrusion (Fig. 4i). Here, the ability of a cell to maintain membrane potential under stress indicates the cell's fitness. Epithelial cells need to work collectively to perform most organ functions and maintain a tight monolayer. Thus, cells that might compromise either aspect must be preferentially eliminated. We identify ENaC as the earliest mechanosensor that measures this fitness by probing how well they can withstand the pressure as cells jostle for position. Epithelial cells with sufficient ATP can withstand crowding when ENaC activates Na$^+$ ingress by using the Na$^+$/K$^+$ ATPase to rebalance their membrane potential. However, crowded cells challenged by ENaC activation with insufficient ATP levels to repolarize the membranes through the Na$^+$/K$^+$ ATPase will fail to pump out Na$^+$. This intercellular increase in Na$^+$ activates K$_v$1.1 and K$_v$1.2, causing cell shrinkage through water egress. If the cell shrinks more that 17%, it will then activate extrusion (Fig. 4i).

## Discussion

This model highlights energy sufficiency as a driving force in determining which cells will die, analogous to mitochondrial permeabilization acting as a commitment step during the canonical apoptosis pathway[35,36]. Yet, during homeostatic extrusion, cells are eliminated while they are still alive and can repopulate if given a new substrate[2]. This mechanism therefore selects for the cells with comparably less energy than their neighbours, rather than no energy. Although neurons can use as much as 70% of their total ATP maintaining the membrane potential[37], epithelial cells are thought to use about 25% of their ATP to do so[32]. Thus, this mechanism could amplify deflections in energy to trigger a cell's exit while it still has sufficient energy to extrude. It is not clear how cell shrinkage activates the downstream extrusion pathway. One idea is that it sets a deeper threshold for crowding to activate Piezo1, damping the constant buffeting of cells among each other. Notably, a recent study found that imposing a direct current from apical to basal side of a monolayer causes wide-scale contraction and induction of extrusion, whereas applying a current that reinforces normal polarity (from basal to apical) enhances and even repairs impaired epithelial cell–cell junctions[38], supporting our model.

Our study identifies ATP, membrane potential and hydrodynamic regulators as the earliest steps controlling homeostatic, crowding-induced cell extrusion—a central driver of epithelial cell death. As energy is key for selecting which cells are eliminated, our work may offer insights into how this process may go awry in metabolic diseases such as diabetes and cancer. Further work will need to determine how cell shrinkage activates Piezo1 to trigger S1P signalling vesicles, essential to extrusion[1,3,4]. Alternatively, the membrane potential differential between a cell targeted for extrusion and its neighbours could set up a directional current, which has been shown to activate migration during wound healing[39] and organ patterning[40].

Our findings also reveal the importance of other ion channels in regulating extrusion, and it will be important to define what roles they have. For example, known drivers of cystic fibrosis include malfunctions of ENaC and CFTR[41–43], and mutations and misregulation of ENaC, K$_v$1.1 and K$_v$1.2 are indicators of poor prognosis in a variety of cancers[44,45]. Moreover, aquaporins may work in conjunction with the K$^+$ and Cl$^-$ channels that we identify; further studies will need to define which, if any, have a role in this process. The channels identified here may be specific for kidney and lung epithelia, yet similar channels may work analogously in different tissues[46] to tune cell size and turnover, depending on organ function and shape. As epithelial cell extrusion is fundamental for epithelial cell turnover, misregulation of extrusion may contribute to disease aetiologies arising from channelopathies.

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

# Methods

## Cell culture

MDCKII cells from the European Collection of Authenticated Cell Cultures (ECACC) operated by Public Health England, catalogue number 00062107, lot 19G037, (tested for *Mycoplasma*; authenticated before receipt) were cultured in Dulbecco's minimum essential medium (DMEM) high glucose with 10% FBS (Thermo Fisher Scientific) and 100 µg ml$^{-1}$ penicillin–streptomycin (Invitrogen) at 5% $CO_2$, 37 °C.

**Animal models and PCLSs.** All animals were housed under specific pathogen-free conditions and cared for in accordance with the UK Home Office Animals (Scientific Procedures) Act of 1986 and the guidelines set by the Institutional Committees on Animal Welfare, project licence P68983265. Animal experiments received approval from the Ethical Review Process Committee at King's College London and were conducted under a Home Office licence in the UK.

Ex vivo lung slices were obtained from male and female mice (*B6N. 219S6(Cg)-Scgb1a1^{tm1(cre/ERT)Blh}/J*) from 7 to 17 weeks of age. In brief, mice were humanely euthanized by injectable anaesthetic overdose followed by exsanguination through the femoral artery. The chest cavity was opened and the trachea was carefully exposed, where a small incision was made to accommodate the insertion of a 20Gx1.25 needle in a canula (SURFLO I.V. catheter). The lungs were then inflated with 2% low melting agarose (Thermo Fisher Scientific, BP1360) prepared in HBSS+ (Gibco, 14025). Then, lungs, along with the heart and trachea, were excised, washed in PBS and the lobes separated. Individual lobes were then embedded in 4% low-melting-point agarose and solidified on ice. Slices (thickness, 200 µm) were cut on a Leica VT1200S vibratome, washed and incubated in DMEM/F-12 medium supplemented with 10% FBS and antibiotics overnight (37 °C, 5% $CO_2$). The ex vivo lung slices were imaged 24 h after dissection. Sections were experimentally treated under conditions described below and analysed blinded.

**Osmolarity solutions.** To test which osmolarities could drive LCE, we treated MDCKII cells or ex vivo precision-cut lung slices (both cultured in DMEM) with increasing amounts of D-mannitol (Sigma-Aldrich, M4125-1kg) or nuclease-free water (Ambion, AM9937) to create hyper or hypotonic medium, respectively. Initial DMEM osmolarities were measured using a freezing-point osmometer (Gonotec, Osmomat 3000), ranged from 334 to 368 mOsm kg$^{-1}$, and were tested biweekly and each time a new batch was prepared.

**Immunostaining.** Cells were fixed with 4% formaldehyde in PBS at room temperature for 20 min, rinsed three times in PBS, permeabilized for 5 min in PBS containing 0.5% Triton X-100 and blocked for 10 min in AbDil (PBS + 5% BSA). The coverslips were then incubated in primary antibody (in PBS + 1% BSA) overnight at 4 °C, washed three times with PBS and incubated in secondary fluorescently conjugated antibodies. All antibodies were used at a dilution of 1:200 unless otherwise specified: rabbit Piezo1 (Novus, NBP1-78446), mouse S1P (Santa Cruz, sc-48356), rabbit KCNA1 (Alomone Labs, APC-161), rabbit KCNA2 (Alomone Labs, APC-010), rabbit LRRC8A (Alomone labs, AAC-001), mouse ZO1 (Invitrogen, 33-9100), rabbit ENaC antibodies SCNNA1 (Invitrogen, PA1-920A), SCNNB1 (Invitrogen, PA5-28909) and SCNNG1 (Invitrogen PA5-77797). Alexa Fluor 488, 568 and 647 goat anti-mouse and anti-rabbit IgG were used as secondary antibodies (Invitrogen). F-actin was stained using either conjugated 488 or 568 phalloidin (66 µM) at 1:500 and DNA with 1 µg ml$^{-1}$ DAPI (Thermo Fisher Scientific) in all fixed-cell experiments.

For PLCSs, untreated slices were fixed with 4% paraformaldehyde or after 30 min or 2 h following various treatments, then blocked for 1 h in AbDil at room temperature and then incubated at 4 °C overnight in 1:100 primary rabbit anti-E-cadherin antibody (24E10, Cell Signaling 3195) in AbDil. After three 30-min washes in PBS + 0.5% Triton X-100, the slices were incubated again at 4 °C overnight with secondary antibodies (1:100 Alexa Fluor 488 goat anti-rabbit at (Thermo Fisher Scientific, A11008) + 1:100 Alexa Fluor 568 Phalloidin (Thermo Fisher Scientific, A12380). For live imaging, BioTracker ATP-Red live-cell dye 1:200 (Sigma-Aldrich SCT045) was incubated at 37 °C for 30 min before imaging.

**Experiment and quantification methods. Extrusion.** Extrusions from time-lapse phase videos of MDCKII cells and PCLSs were quantified by identifying cells that were eliminated from the monolayer or tissue through classical squeezing out from the surrounding monolayer. These cells were then followed backwards in time to quantify cell shrinkage, compared with initiation of extrusion. By contrast, cells that round up, divide and reincorporate into the monolayer were scored as mitoses. The cells that were already eliminated by extrusion at the beginning of filming were excluded from our quantifications.

**QPI analysis.** QPI acquisition relies on having nearby cell-free areas to measure cell mass. To achieve this, we grew monolayers on small patterned circles within a dish by adhering a silicone laser-cut 100 micromesh disk (Micromesh Array, MMA-0500-100-08-01) onto a non-tissue-culture-treated 35 mm dish (Ibidi, 81151). The dishes were plasma treated with the mesh in place using a chamber (Harrick Plasma, Cleaner PD-32G) applied in a vacuum (Agilent Technologies, IDP-3 dry scroll vacuum pump) for 10 min to create cell growth in a pattern required for quantification (Extended Data Fig. 1a). Immediately after plasma treatment, the silicone mesh was aseptically removed and MDCKII cells were seeded at a density of 128,000 cells per well in a 35 mm microscopy imaging dish (Ibidi, 81151) and incubated at 37 °C for 6 h in DMEM. Before filming, excess cells were removed from unpatterned areas by gently washing twice with DMEM and growing another 48 h.

To image, cells were placed in an on-stage incubator and islands showing the entire cell island boundary and encompassing empty space were filmed on a QPI microscope. A minimum of two images of empty space was used for background correction. Images were acquired every 2 min for 10 h at 37 °C, 5% $CO_2$ and 88% humidity.

The dry mass was then calculated by subtracting the reference images from cells within the island to correct for background. The background was adjusted by subtracting the average phase shift of the empty space from the whole field of view including areas covered by cells. By subtracting the background, we could acquire the island and cell phase shift per image. The phase shift was converted to dry mass using previously established methods[11,47] in MATLAB (v.R2022a). Extrusions were quantified from 12 separate islands.

**$Ca^{2+}$ and $K^+$ quantification.** MDCKII cells were plated at a density of 28,000 cells per well in a 35 mm microscopy imaging dish (Ibidi, 81156) and incubated at 37 °C overnight or until 60% confluent. Once 60% confluent, cells were transfected according to the manufacturer's protocol with Lipofectamine 3000 (Thermo Fisher Scientific, L3000001) with genetically encoded $Ca^{2+}$ indicator GECO (Addgene, CMV-G_GECO1.0, 32447)[48] or BLINK2 (Addgene, pDONR-BLINK2, 117075) for 18 h. Transfection medium was removed, and cells were rinsed twice with PBS and incubated in DMEM + FBS until cell-to-cell junctions were mature (72 h). To image, cells were stained with Hoechst (Invitrogen, 1:1,000) in PBS for 10 min at 37 °C, washed twice with PBS and then incubated in DMEM medium in an enclosed incubated stage at 37 °C with 5% $CO_2$ (Oko labs). For capturing $Ca^{2+}$ changes, time-lapse images were captured every 10 s using a spinning-disk microscope (Nikon, Ti2) for up to 10 h. Images were analysed using a threshold macro (Nikon Elements AR, 5.41.02) to quantify the $Ca^{2+}$ fluorescence level changes of cells over time that extrude.

The genetically encoded $K^+$ indicator is optogenetically stimulated by blue light to open $K^+$ channels. BLINK2 was activated by selecting the cell as a region of interest (ROI) in Nikon elements and then using a GalvoXY 405 laser at 55% for 300 ms, as any higher caused immediate cell death.

**Volume quantification.** MDCKII cells were plated at a density of 28,000 cells per well in a 35 mm microscopy imaging dish (Ibidi, 81156) and incubated at 37 °C overnight or until 60% confluent. Once 60% confluent, cells were either transfected according to the manufacturer's protocol with Lipofectamine 3000 (Thermo Fisher Scientific, L3000001) with cytoplasmic GFP plasmid (Addgene, pEGFP-N1) for 18 h or incubated when mature (72 h) with Calcein-AM dye (Thermo Fisher Scientific, C1430; 10 μM). Transfection medium was removed, and cells were rinsed twice with PBS and incubated in full DMEM until cell-to-cell junctions were mature (72 h). To image, cells were stained with Deep Red Cell Mask (Thermo Fisher Scientific, 1.5:1,000) for 30 min and Hoechst (Invitrogen, 1:1,000) in PBS for 10 min at 37 °C, washed twice with PBS and incubated in DMEM medium in an enclosed incubated stage at 37 °C with 5% $CO_2$ (Oko labs) and 0.4 μm *z* slices were captured every 10 s using a spinning-disk confocal microscope (Nikon, Ti2) for up to 10 h.

To quantify the volume changes of cells expressing cytoplasmic GFP, images were analysed using a threshold macro (Nikon Elements AR, v.5.41.02) with cell mask membrane-stained boundaries to highlight extrusions. The volume data were normalized to the baseline volume before notable junctional changes or extrusion occurs in Excel (Microsoft) and graphed and analysed using GraphPad Prism v.9.4.1.

To quantify cell shrinkage based on solute content, Calcein-AM fluorescence emission was quantified, whereby decreased fluorescence occurs with cell shrinkage. Calcein fluorescence changes were captured during homeostatic shrinkage, hypertonic induced shrinkage, and in response to ion channel inhibitors and hypertonic treatment. Here, fluorescence data were normalized for each cell to its baseline before homeostatic and or hypertonic induced shrinkage in Excel (Microsoft) and graphed and analysed using GraphPad Prism v.9.4.1.

**Serial osmolarity treatment.** A total of $0.53 \times 10^5$ MDCKII cells was seeded on glass round coverslips (22 × 55 mm; Academy, 400-05-21) and grown to confluence (-100 h). Epithelial monolayers were incubated with isotonic DMEM for 10 min, before treating for 10 min with increasing concentrations of hyper or hypotonic medium to induce cell shrinking or swelling, respectively, then with isotonic DMEM for 120 min. Cells were either filmed (see below) or fixed and stained to quantify extrusions. Experiment cell densities were analysed with bright spots macro in NIS Elements General Analysis (Nikon Elements AR, v.5.41.02) using DNA staining to determine cell density per field and phalloidin and DNA to identify extrusions.

**Live hypertonic shock.** For live imaging following hypertonic shock, MDCKII cells were plated on an 8-well slide (Ibidi, IB-80801) at a seeding density of 10,000 cells per well and incubated for about 72 h until monolayers were confluent with mature cell–cell adhesions. Cells were stained with Hoechst (1:1,000) in PBS for 10 min at 37 °C, washed twice with PBS and incubated in isotonic DMEM medium (with or without inhibitors), placed in a microscope stage incubator (37 °C, 5% $CO_2$, Okolabs) and imaged every 10 s for 2.5 h using a widefield microscope (Nikon, Ti2 specifications are provided below). For live cleaved caspase 3 staining, cells were incubated according to the manufacturer's instructions (1:200, Incucyte caspase-3/7 dyes) before imaging. All of the experiments consisted of three phases: (1) baseline: 0–10 min, during which the cells are incubated in isotonic medium (with or without inhibitors (Supplementary Table 1) or siRNA knockdown (Extended Data Fig. 3)); (2) hypertonic challenge: 10–20 min, during which imaging is paused while isotonic medium is replaced with 20% hypertonic medium with or without inhibitors before imaging is rapidly resumed to capture shrinkage; (3) effects on extrusion: from 20 min to the end of imaging, during which imaging is paused while replacing 20% hypertonic medium with or without inhibitors is replaced with isotonic medium with or without inhibitors and time-lapse phase imaging is resumed to capture. Thus, contractility appears to suppress cell shrinkage over the next 2 h.

PCLSs were imaged to establish the baseline conditions before treatments. For hypertonic challenge, the slices were incubated in 40% hypertonic solution for 20 min, then transferred to isotonic phase imaging medium for live imaging. The drug treatment effect was imaged right after incubating PCLSs in isotonic medium treated with a combination of oligomycin A and oxamate. The experiments were then quantified for extrusion rates per 1,000 or 10,000 cells over time-lapse videos identified using phase microscopy or for the percentage of shrinkage using lightning assays described below.

**siRNA knockdown.** Four-siRNA smart pools (Horizon Discovery, L-006210-00-0010 ($K_v1.1$), L-006212-00-0010 ($K_v1.2$), L-026211-01-0010 (SWELL1), L-006504-00-0010 (ENaCα), L-006505-00-0010 (ENaCβ), L-006507-01-0010 (ENaCγ) or D-001810-01-20 (non-targeting control)) were prepared in DNase/RNase-free water to a 100 μM stock. Then, 28,000 MDCKII cells were seeded in a 6-well plate for quantitative PCR (qPCR) analysis (Thermo Fisher Scientific, 140675), at 10,000 cells per well of an 8-well slide (Ibidi, IB-80801) for live-cell imaging, or with 53,000 cells per 24-well dish with coverslips (Thermo Fisher Scientific, 142475) for extrusion quantification, and grown overnight until 60% confluent. Cells were then transfected using the RNAi Max kit (Thermo Fisher Scientific, 13778150) and 1 μM siRNA for 24 h before replacing with fresh DMEM for 48 h. Cell knockdowns plated for qPCR with reverse transcription (RT–qPCR) analyses were lysed for RNA extraction using the RNAeasy kit (Qiagen, 74104) according to the manufacturer's instructions. RNA (1 μg) was purified with 1 μl of 10× DNase I reaction buffer, 1 μl of DNase I amplification and RNase-free water in a final volume of 10 μl. The samples were incubated for 10 min at 37 °C, then the reaction was deactivated with 1 μl of 0.5 M EDTA for 10 min at 75 °C. The samples were stored at −20 °C or directly processed by RT–qPCR using the Brilliant III Ultra-Fast SYBR Green QRT-PCR Master Mix (Agilent Technologies), using primers designed with SnapGene (v.6.1.1) and produced by Sigma-Aldrich (Extended Data Fig. 3b). Reactions were analysed using the ViiA 7 Real-Time PCR System (Thermo Fisher Scientific) using the following cycle conditions: 50 °C for 10 min, 95 °C for 3 min, followed by 40 cycles at 95 °C for 15 s and 60 °C for 30 s. Results were normalized to *GAPDH* expression and graphed and statistically analysed using GraphPad Prism v.9.4.1. Extrusion rates were quantified per 1,000 cells using time-lapse phase microscopy and the percentage of shrinkage was determined using the lightning assay as described below.

**Lightning assay.** To expedite analysis of cell shrinkage after modulation of different channels, we used the lightning assay. Regions of interest were cropped from phase-microscopy time-lapse videos, thresholding the phase-bright junctional intensity based on white-light detection before, during and briefly after extrusion. The threshold was set to capture the area changes around the cells in the frames before HES or OICE. The same threshold was applied to all frames of the video until completion of cell extrusion. This same method was used for both single cell and whole regions of crowding in an 85 μm$^2$ (400 px by 400 px) area. Data were then normalized in Microsoft Excel (v.16.67) using an average of 10 frames before lightning and analysing the peak percentage change, and then graphed and statistically analysed using GraphPad Prism v.9.4.1.

**Depolarization.** A total of 128,000 MDCKII cells per 35-mm dish was grown around 72 h to maturity and then stained with $DiBAC_4(3)$ according to the 'Tracking transmembrane voltage using $DiBAC_4(3)$ fluorescent dye (PDF)' protocol (https://ase.tufts.edu/biology/labs/levin/resources/protocols.htm).

Monolayers were then treated with DMSO (vehicle), 4-AP or amiloride, and imaged (phase and GFP settings) every 10 s for a minimum of 2.5 h. PCLSs were time-lapse imaged after incubating with 1:500 DRAQ5 fluorescent probe solution (5 mM, Thermo Fisher Scientific, 62251), 1:500 ATP and 1:1,000 $DiBAC_4(3)$ in HBSS for 30 min at 37 °C according to the manufacturer's instructions. $DiBAC_4(3)$ was refreshed at each medium/treatment change.

All of the images were analysed using Nikon Elements AR (v.5.41.02) using a ROI over any cell that was maintained or extruded. The ROI mean intensity of $DiBAC_4(3)$ in each cell over time was normalized in Excel using 10 baseline frames before the shrink and depolarization

over time was graphed in GraphPad Prism v.9.4.1. Cell counts were then plotted and analysed in Graph Pad Prism v.9.4.1.

**ATP measurements.** ATP levels were followed in both MDCKII cells and PCLSs using ATP-Red or Queen37. In total, 10,000 MDCKII cells were seeded per well of an 8-well slide (Ibidi, IB-80801) and grown to maturity. ATP levels were analysed after transfection (as described above) of the genetically encoded ATP indicator Queen37 (Addgene pN1-QUE37C, 129318) and or stained with ATP-Red live dye. Transfected cells were counterstained with ATP-Red (10 µM) for 30 min at 37 °C and washed twice with PBS, and then incubated in DMEM medium in an enclosed incubated stage at 37 °C with 5% $CO_2$ (Oko labs) and 0.4 µm $z$ slices were captured every 2 min using a spinning-disk microscope (Nikon, Ti2) for up to 3 h. Images were analysed using a threshold macro (Nikon Elements AR, v.5.41.02) to quantify the fluorescent changes of cells expressing Queen37 and ATP to highlight changes before extrusions. The fluorescence data were normalized to the baseline levels before depletion and extrusion in Excel (Microsoft) and graphed and analysed using GraphPad Prism v.9.4.1.

Further time-lapse experiments with mature MDCKII cells grown to confluency were incubated with live with ATP-Red, $DiBAC_4(3)$ (as previously described), or CoroNa green AM (described below in the crowding subsection) with or without the ATP inhibitors oligomycin A or oxamate (Supplementary Table 1), or treated with Piezo1 inhibitor GsMTx4, or supplementing with glucose with addition of DMEM with high glucose.

Moreover, PCLSs were time-lapse imaged incubated with 1:500 DRAQ5 Fluorescent Probe Solution (5 mM, Thermo Fisher Scientific, 62251) and 1:500 ATP-Red in HBSS for 30 min at 37 °C according to the manufacturer's instructions with or without the ATP inhibitors oligomycin A or oxamate (Supplementary Table 1). Imaging was performed in glass-bottom 24-well plates (Ibidi, 82427), with a glass coverslip placed on top to prevent drifting during 4 h imaging sessions at 5 min intervals.

**Crowding.** MDCKII cells were seeded at 128,000 cells per well in a uniaxial stretched (25%) 10 $cm^3$ PDMS chamber (Strexcell, SC-0100) and grown ~72 h to confluence and junctional maturity. Once mature, monolayers were stained with the $Na^+$ indicator CoroNa green AM (Thermo Fisher Scientific, C36676 at 10 µM), ATP-Red live-cell dye at 10 µM and cell mask according to the manufacturer's instructions. The cells were then imaged at homeostatic density or after crowding by releasing monolayers from stretch with or without the ATP inhibitor oligomycin A. Here, fluorescence data were normalized to a previous measurement taken before crowding, or to an earlier timepoint before corresponding homeostatic timepoints that match time of crowding, for each cell. $Na^+$ and ATP fluorescence changes and extrusion rates were quantified in Excel (Microsoft) and graphed and analysed using GraphPad Prism (v.9.4.1).

**Microscopy equipment. QPI.** Time-lapse QPI and brightfield images were collected on the Olympus IX83 inverted microscope (Olympus) using a ×40/0.75 NA objective. The samples were illuminated using red LED light (623 nm, Thorlabs, DC2200) for 120 ms exposure with a QWLSI wavefront sensing camera (Phasics SID4-4MP), driven by Micro Manager open-source microscopy software. The samples were incubated with a stage-top incubator (Okolabs) set at 37 °C temperature with 5% $CO_2$ gas and 95% humidity.

**Widefield imaging.** Time-lapse phase and fluorescence images were captured on the Nikon Eclipse Ti2 system using a Plan Fluor ×20 Ph1 DLL NA = 0.50 objective with a Photometrics Iris 15 16-bit camera and a Cool LED pE-4000 lamp driven by NIS Elements (Nikon, v.5.30.02).

**Spinning-disk microscopy.** Images were captured on the Nikon Eclipse Ti2 system using Plan Fluor ×20 or ×40 0.75 air objectives or Plan Fluor ×60 or ×100 1.40 oil objectives with an iXon 888 Andor 16-bit camera, a Yokogawa CSU-W1 confocal spinning-disk unit and a Toptica photonics laser driven by NIS Elements (Nikon, v.5.21.03). For blue-light optogenetic stimulation, a Galvo-meter XY (Brunker) 405 laser was used with the Ti2 microscope. Cell staining with phalloidin and Hoechst were quantified for extrusions per 1,000 or 10,000 cells using Nikon Elements Software.

## Statistics and reproducibility

For statistical analysis, all experiments were repeated independently on at least three separate days to capture variation in the biological replicates. The minimum sample size was determined according to the standards in the field and based on previously established calculations[49]. This includes graphed data and representative data such as pictures and micrographs. Data were analysed using GraphPad Prism v.9.4.1 statistical software to measure normality with the Shapiro–Wilk test and significance using unpaired and ratio paired $t$-tests (all $t$-tests were performed with two-tailed analysis), two-way ANOVA with Tukey's or Šidák's correction, or one-way ANOVA with Dunnett's or Welch multiple-comparison correction, as described in the figure legends. To reduce bias, we imaged random fields within the middle or crowded areas of glass coverslips for quantification or the centre of an 8-well dish (Ibidi, 80806) for live-cell shrink experiments. We excluded low-density epithelia (fields with less than 2,000 cells) that are not crowded enough to elicit extrusion. Graphs were generated using GraphPad Prism v.9.4.1. Figure layouts and models were created in Adobe Illustrator (v.26.3.1). As much of the data analyses were done by the person running the experiments, it was not possible to blind all analysis. However, findings were confirmed with other author investigators who were blinded to the analysis. Further, most analyses and data collection depended on software capture of predetermined set parameters of fluorescence or white light from cell dyes and/or genetically encoded probes or phase microscopy. As the parameters were set on the basis of controls, the data collection was semi-automated and therefore not collected without bias.

## Reporting summary

Further information on research design is available in the Nature Portfolio Reporting Summary linked to this article.

## Data availability

Owing to their large size (total of roughly 24 TB), raw microscopy data could not be made available but can be obtained from the corresponding author on request. Source data are provided with this paper.

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

**Acknowledgements** We thank M. Redd for his in-depth questioning that led to revisions to the models and the manuscript; members of the Rosenblatt laboratory for comments on our manuscript; and N. Lane and J. Ashmore for insights on this study. J.R. is the recipient of National Institute of Health R01GM102169, a Howard Hughes Faculty Scholar Award (55108560), a Cancer Research UK Programme Grant (DRCNPG-May21\100007), an Academy of Medical Sciences Professorship (APR2\1007) and a Wellcome Trust Investigator Award (221908/Z/20/Z). C.P.-P. is the recipient of a long-term fellowship (LT000654/2019-L) from the Human Frontier Science Program organization and a Marie Skłodowska-Curie Fellowship (898067) from the European Union's Horizon 2020 research and innovation programme. This work was supported by the Office of the Assistant Secretary of Defense for Health Affairs through the Breast Cancer Research Program under Award Number W81XWH1910065 (T.A.Z.).

**Author contributions** J.R. and S.J.M. designed experiments, interpreted, analysed data and wrote the manuscript. C.P.-P. designed hypertonic and ion channel experiments, interpreted data and offered important guidance throughout the paper. A.T. designed, performed and analysed mouse lung experiments. T.A.Z. designed and analysed QPI experiments. All of the authors edited the manuscript.

**Competing interests** The authors declare no competing interests.

**Additional information**
**Correspondence and requests for materials** should be addressed to Jody Rosenblatt.

## A  MICROPATTERN MONOLAYER

## B  QPI MONOLAYER

EXTRUSION

0:46  1:02  1:14  1:16  1:18

DIVISION

0:00  0:06  0:12  0:18  0:38

0 ▬ 3 pg/µm²

## C

dividing cell

extruding cell

DRY MASS (pg)

350
300
250
200

-60  -50  -40  -30  -20  -10  0

TIME RELATIVE TO EXTRUSION/DIVISION (MIN)

## D  MDCK II

BEFORE  -60′  SHRINK  0′  EXTRUSION  38′

-60′  0′  38′

Phase GECO-Ca²⁺

## E

Ca²⁺ INTENSITY

600
550
500
450
400
350

SHRINK

14:12:00  14:23:59  14:36:00  14:47:59  15:00:00

TIME (HR:MIN:SEC)

**Extended Data Fig. 1** | See next page for caption.

**Extended Data Fig. 1 | Quantitative phase imaging (QPI) shows unaltered dry mass in cells that extrude.** A, As QPI requires a reference area with no cells throughout imaging, we grew MDCKII monolayers on confined 100 μm islands within a 35 mm dish by plasma treating patterned areas, as shown in example. Scale bar=100 μm. B, Representative QPI stills from 31 cells from 12 QPI islands in 3 experiments (from Supplemental video 1), showing potential dry mass differences before cell extrusion, where blue is low cell mass and yellow is higher mass increases only after the cell contracts and extrudes (h:mm). By contrast, another cell, in red box, increases mass before dividing, as expected. Scale bar = 50 μm inset box scale bar = 20 μm. C, Graph showing the dry mass of the extruding or dividing cell sixty minutes before each event. D, Representative timelapse images of cells prior to extrusion labelled in red with genetically encoded calcium indicator GECO (red) and phase contrast, scale bar = 5 μm. E, Representative fluorescent trace over time of calcium in cell that extrude. All n's are independent two tailed experiments.

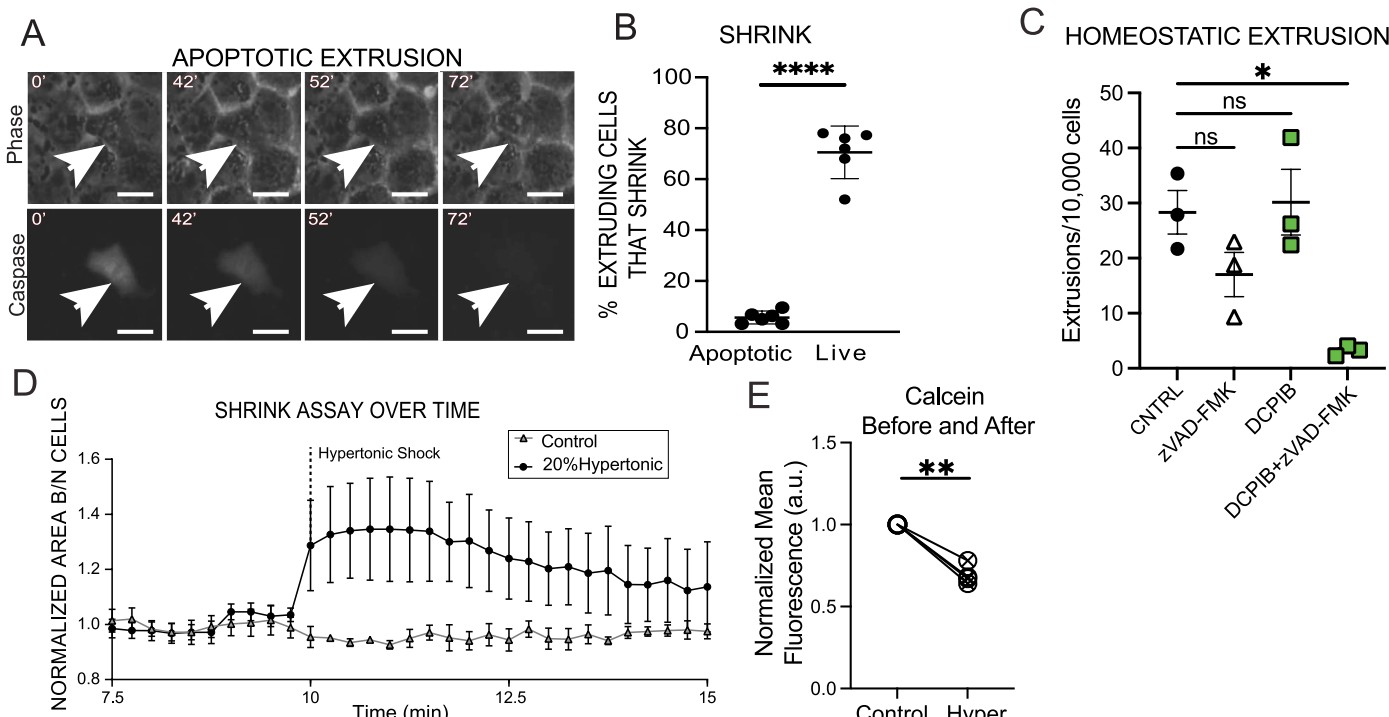

**Extended Data Fig. 2 | Shrink induces live cell extrusion.** A, Stills from a phase (top) timelapse with fluorescent apoptotic marker (bottom) before apoptotic extrusion; scale bar=10 μm. Quantified in (B) as percentage of cells that shrink before extruding. Data represented as mean ± SEM; n = 5; ****P < 0.0001 by an unpaired T-test of cell extrusion type. C, Mean number of extrusions ±SEM in monolayers treated with SWELL1 inhibitor alone or in combination zVAD-FMK; n = 3; *P = 0.0104 from one-way ANOVA with Sidak's multiple comparisons test. D, Adapted lightning assay shows area around cells increases with introduction of hypertonic media over time. E, Normalized mean calcein fluorescence before (control isotonic conditions) and after hypertonic conditions, n = 4; where **P = 0.0021 values are from a paired T-test. All n's are independent two tailed experiments.

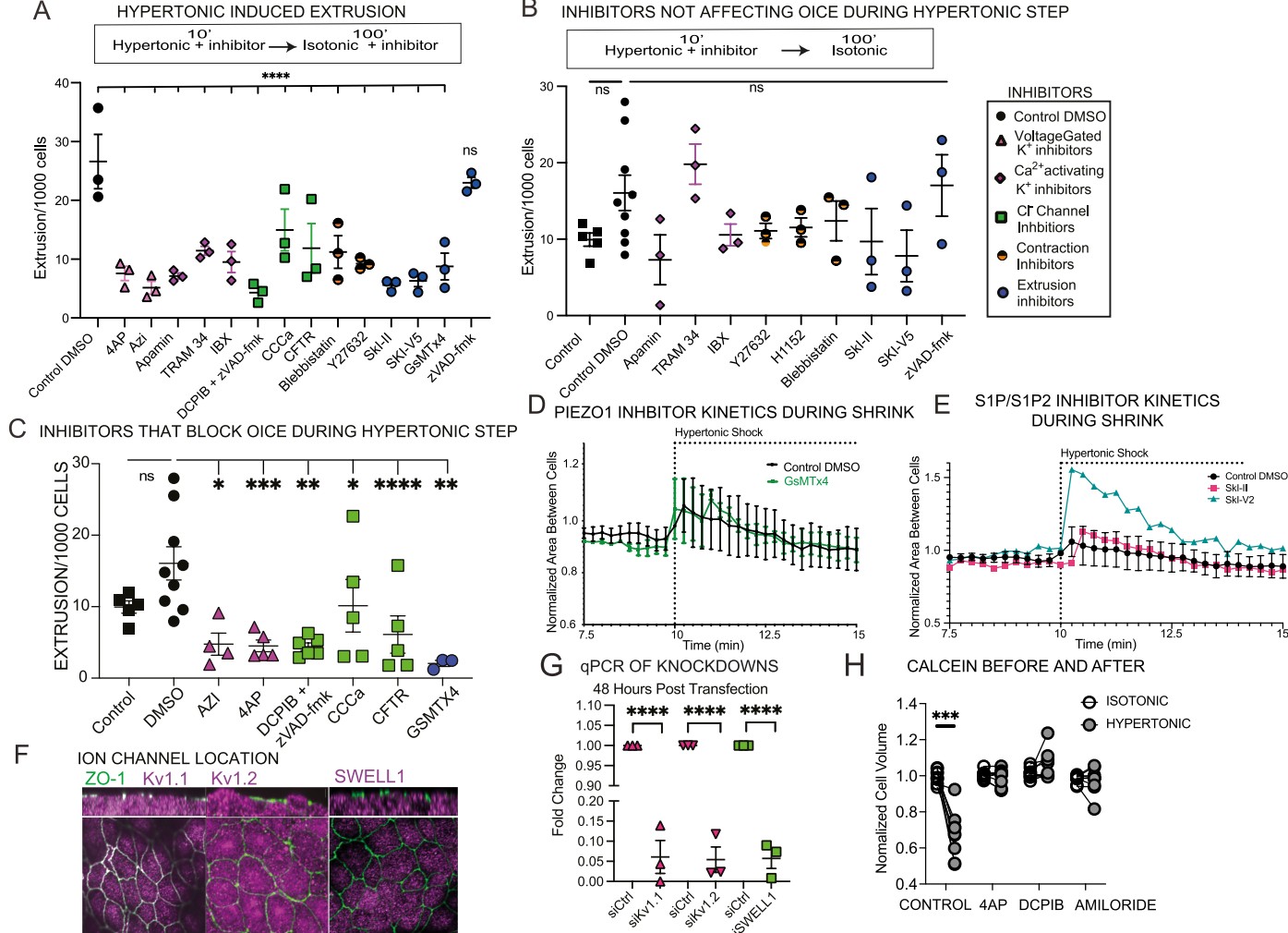

**Extended Data Fig. 3 | Ion channel inhibitor analysis on extrusion and shrink.** A, Mean extrusion rate ±SEM in monolayers pretreated with inhibitors and during both hypertonic treatment and return to isotonic media; n = 3; ****P = 0.0001 by one way ANOVA with Dunnett's multiple comparisons test. B, Mean extrusion rate ±SEM pre-treated with inhibitors but only continued during hypertonic treatment, boxes outlined above graphs. N = 3 experiments where NS is from two-way ANOVA with Dunnett's multiple comparisons test. Inhibitor key for both A and B describes ion channel family inhibitor target by assigned colour and icon. C, Mean extrusion rate ±SEM from monolayers treated with inhibitors only during the 10' 20% hypertonic treatment n = 3; *P = 0.0142 (AZI), ***P = 0.0003 (4AP), **P = 0.003 (DCPIB), *P = 0.0134 (CCCa),

****P < 0.0001(CFTR), **P = 0.0011 (GsMTx4) from two-way ANOVA with Dunnett's multiple comparisons test. Representative "Lightning assay" where increased space around the cells indicates cell shrinkage in the presences of: (D) SAC inhibitor GsMTx4, or (E) S1P/S1P₂ inhibitors compared to DMSO controls before and during hypertonic media incubation (mins). F, Confocal representative projections and XZ images of Kv1.1/1.2, or SWELL1 (magenta) with apical tricellular junction protein ZO-1 (green). XY Scale bar=20 μm; n = 3. G, Scatter plots show fold changes (2^−ΔΔCt) at 48 h post transfection; the first shows n = 3; ****P = 0.0001 from an unpaired T-Test. H, Normalized mean cell volume ±SEM before and after experimentally inducing shrink, n = 6; ***P = 0.0004 from ratio paired two-tailed T-test. All n's are independent two-tailed experiments.

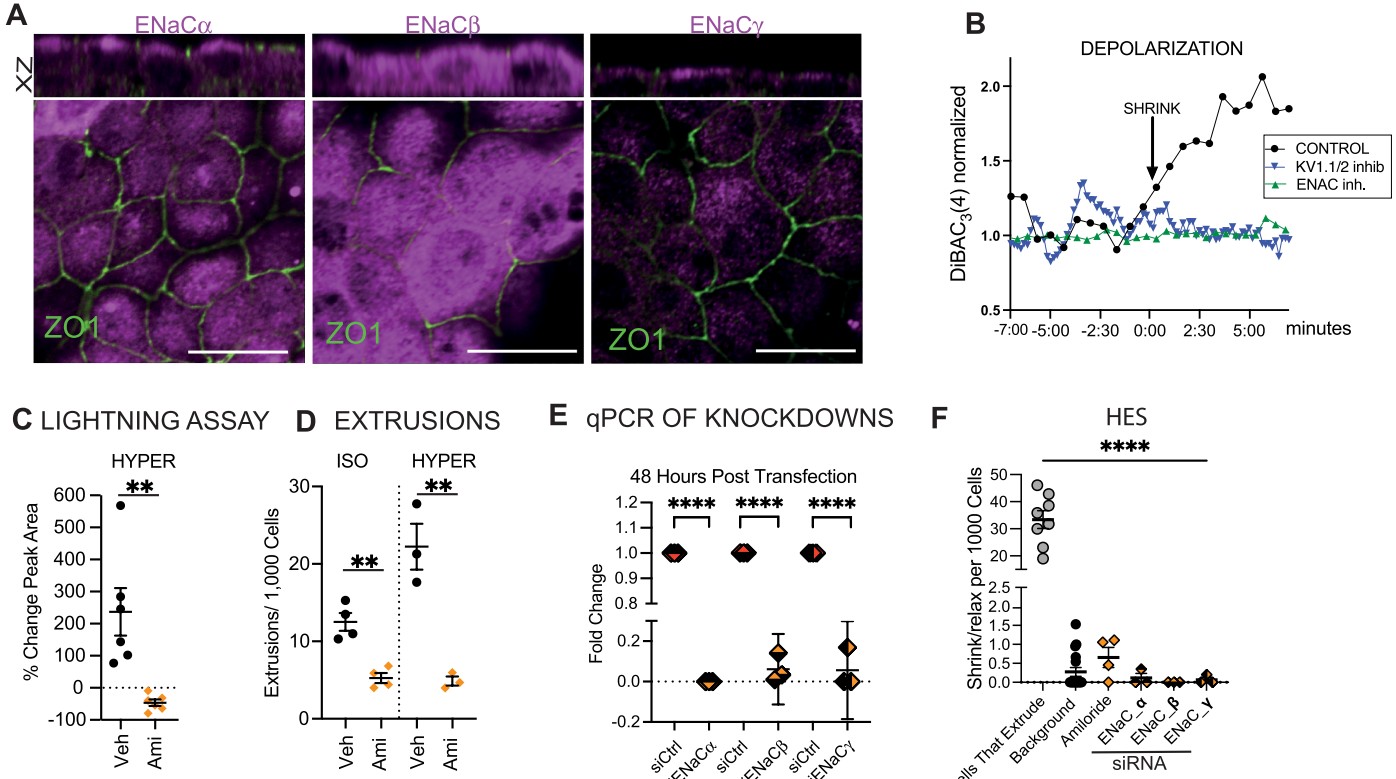

**Extended Data Fig. 4 | ENaC localization and role in OICE and HES.** A, Confocal representative XY projections and XZ slices of ENaC α, β, or γ (magenta) localization with respect to apical tricellular junction protein ZO-1 (green). XY Scale bar = 20 μm; n = 3. B, Representative depolarization of cells over time (mm:ss) with DMSO control, Kv1.1/1.2 inhibitor (4-AP) or ENaC inhibitor (amiloride). C, Lightning assay as mean % peak area change ±SEM of cells with amiloride during 20% hypertonic challenge, n = 6, **P = 0.0036 from an unpaired t-test. D, Mean extrusion rate ±SEM with amiloride compared to DMSO controls.

n = 4 isotonic media treatments and n = 3 for hypertonic challenges, **P < 0.0015 (iso), **P = 0.0045 (hyper) by an unpaired t-test. E, Scatter plots show fold changes ($2^{-\Delta\Delta Ct}$) at 48 h post transfection; the first shows n = 3; ****P < 0.0001 from an unpaired T-Test. F, Mean cell shrinkage rate ±SEM during steady state turnover ±ENAC inhibition or ENaC-α, β or γ siRNA, compared to cells that shrink before extruding; n = 3 ****P < 0.0001 from two-way ANOVA with Dunnett's multiple comparisons test. All n's are independent two-tailed experiments.

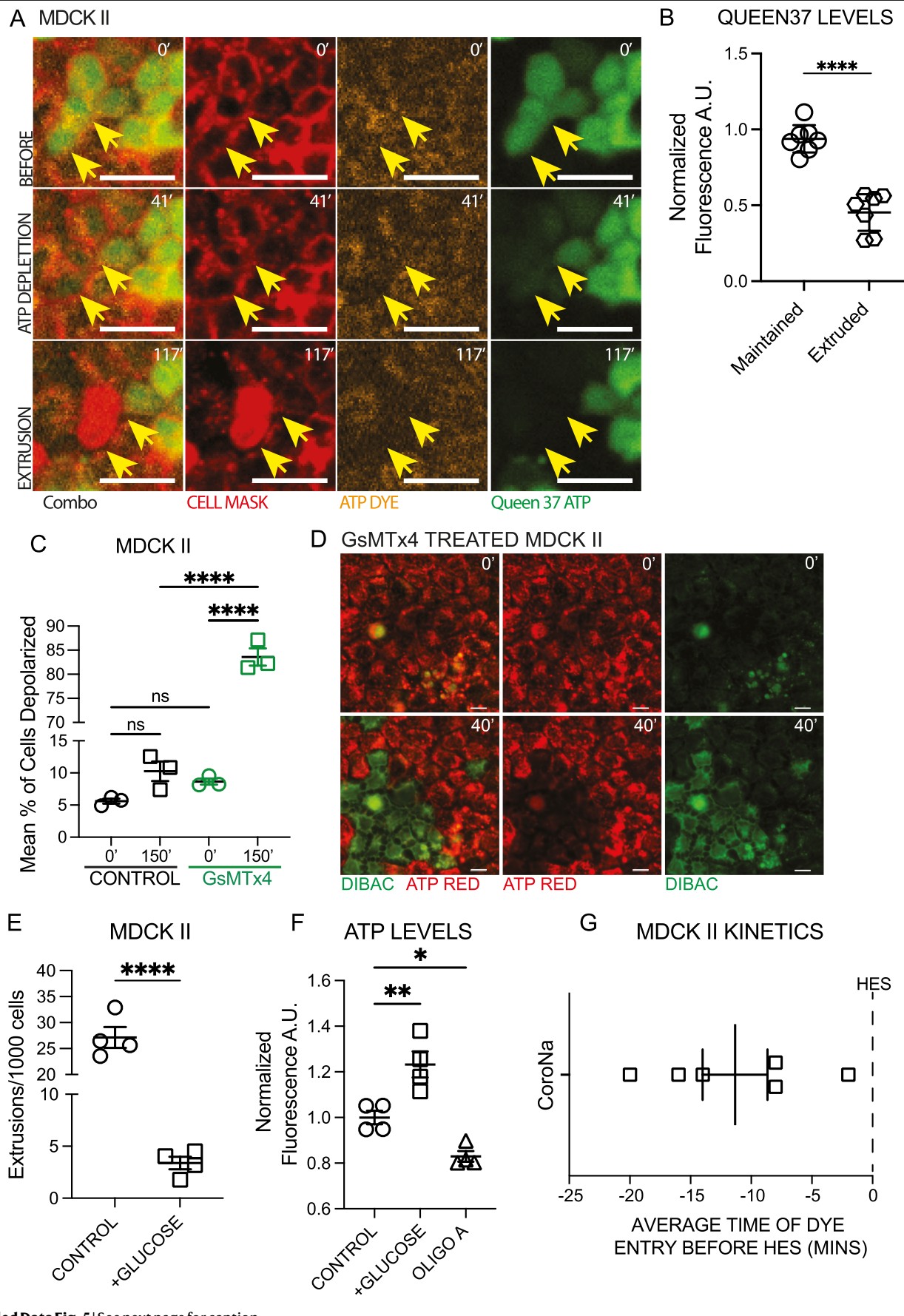

**Extended Data Fig. 5 |** See next page for caption.

**Extended Data Fig. 5 | Further analysis of ENaC and voltage.** A, Confocal representative XY images of MDCKII cells transfected with ATP indicator Queen37, ATP Red live dye and cell membrane marker cell mask (red) before and after extrusion. Yellow arrows point out the cells which will undergo extrusion; Scale bar=25 μm. Quantified as (B), mean normalized fluorescence of Queen37 genetically encoded marker in cells that extrude to those that do not; n = 7, ****P ≤ 0.0001 by unpaired T-Test. C, Percentage of depolarized cells ±SEM over time ± GsMTx4, n = 3; ****P < 0.0001 from an two-way ANOVA with Sidak multiple corrections. D, Widefield representative XY images of MDCKII cells stained with ATP Red live dye and DiBAC depolarization indicator treated with or without GsMTX4 over time; scale bars =10 μm; n = 3. E, Mean extrusion rate ±SEM ±glucose, n = 4; ****P < 0.0001 from student T-Test. F, Average time CoroNa AM (sodium dye) or DiBAC (depolarization dye) enters cells before HES (dotted line), n = 4 (5 cells), ****P < 0.0001 from student T-Test. G. Average time ±SEM sodium dye (CoroNa) remains in the cytoplasm before HES; n = 3. All n's are independent two-tailed experiments.

# Reporting Summary

## Statistics

For all statistical analyses, confirm that the following items are present in the figure legend, table legend, main text, or Methods section.

| n/a | Confirmed | |
|---|---|---|
| ☐ | ☒ | The exact sample size (*n*) for each experimental group/condition, given as a discrete number and unit of measurement |
| ☐ | ☒ | A statement on whether measurements were taken from distinct samples or whether the same sample was measured repeatedly |
| ☐ | ☒ | The statistical test(s) used AND whether they are one- or two-sided<br>*Only common tests should be described solely by name; describe more complex techniques in the Methods section.* |
| ☐ | ☒ | A description of all covariates tested |
| ☐ | ☒ | A description of any assumptions or corrections, such as tests of normality and adjustment for multiple comparisons |
| ☐ | ☒ | A full description of the statistical parameters including central tendency (e.g. means) or other basic estimates (e.g. regression coefficient) AND variation (e.g. standard deviation) or associated estimates of uncertainty (e.g. confidence intervals) |
| ☐ | ☒ | For null hypothesis testing, the test statistic (e.g. *F*, *t*, *r*) with confidence intervals, effect sizes, degrees of freedom and *P* value noted<br>*Give P values as exact values whenever suitable.* |
| ☒ | ☐ | For Bayesian analysis, information on the choice of priors and Markov chain Monte Carlo settings |
| ☐ | ☒ | For hierarchical and complex designs, identification of the appropriate level for tests and full reporting of outcomes |
| ☐ | ☒ | Estimates of effect sizes (e.g. Cohen's *d*, Pearson's *r*), indicating how they were calculated |

*Our web collection on statistics for biologists contains articles on many of the points above.*

## Software and code

Policy information about availability of computer code

| Data collection | Commercial NIS Elements imaging Analysis Explorer package software version 5.41.02 Specifically the analysis explorer package was used to capture maximum IP (for z stack data to be analysed)  by florescent channel  then background estimates and mean intensity per entire image was captured. Further, the max intensity was used oh phase image to caputre HES and OICE raw data.<br>Excel (microscoft 16.89.1 ) Matlab version R2022a. |
|---|---|
| Data analysis | Fiji {ImageJ l.52p), Excel (microscoft 16.89.1 ), Commercial Graph Pad Prism software  version 9.4.1, and Matlab version R2022a |

For manuscripts utilizing custom algorithms or software that are central to the research but not yet described in published literature, software must be made available to editors and reviewers. We strongly encourage code deposition in a community repository (e.g. GitHub). See the Nature Portfolio guidelines for submitting code & software for further information.

## Data

Policy information about availability of data

All manuscripts must include a data availability statement. This statement should provide the following information, where applicable:
- Accession codes, unique identifiers, or web links for publicly available datasets
- A description of any restrictions on data availability
- For clinical datasets or third party data, please ensure that the statement adheres to our policy

> Due to their large size (total of roughly 24TB), raw microscopy data could not be made available but can be obtained from the corresponding author upon request. Source data are provided with this paper

## Research involving human participants, their data, or biological material

Policy information about studies with human participants or human data. See also policy information about sex, gender (identity/presentation), and sexual orientation and race, ethnicity and racism.

| | |
|---|---|
| Reporting on sex and gender | N/A |
| Reporting on race, ethnicity, or other socially relevant groupings | N/A |
| Population characteristics | N/A |
| Recruitment | N/A |
| Ethics oversight | N/A |

Note that full information on the approval of the study protocol must also be provided in the manuscript.

# Field-specific reporting

Please select the one below that is the best fit for your research. If you are not sure, read the appropriate sections before making your selection.

☒ Life sciences          ☐ Behavioural & social sciences          ☐ Ecological, evolutionary & environmental sciences

For a reference copy of the document with all sections, see nature.com/documents/nr-reporting-summary-flat.pdf

# Life sciences study design

All studies must disclose on these points even when the disclosure is negative.

| | |
|---|---|
| Sample size | The minimum sample size was determined according to the standards in the field and based on these calculations: Charan, Jaykaran, and N D Kantharia. "How to calculate sample size in animal studies?." Journal of pharmacology & pharmacotherapeutics vol. 4,4 (2013): 303-6. |
| Data exclusions | The only data exclusions were if MDCKII cellular monolayers were not dense enough to cause extrusion. Since extrusion is a mechanical regulated process of elimination areas or total cell counts were required to meet a density of ~60 cells per 100mm squared. Below this determined that not enough selective pressure lead to extrusion rates as previously described in published work. |
| Replication | Data was replicated minimum of 3 times by repeating the experiments on different days , in cell culture and different passage numbers. Further key experiments were repeated three times in an animal model using invivo mouse lung slices. All replications were successful. |
| Randomization | Extrusion, HES and OICE did occur in dense regions of the monolayers and since that was the mechanism being elucidated which were randomly choosen for analysis. The exception is when assessing the background cells and following extrusion. Cell in this case were identified for extrusion then followed back in time to determine the change in dyes and or phase microscopy changes. |
| Blinding | As much of the data analyses was done by the person running the experiments it was not possible to blind all analysis. However findings were confirmed with other author investigators blinded. Further most analyses and data collection depended on software capture of predetermined set parameters of florescence or white light from cell dyes and or genetically encoded probes or phase microscopy. Since the parameters were set based on controls the data collection was semi-automated and thus not collected with out bias. |

# Reporting for specific materials, systems and methods

We require information from authors about some types of materials, experimental systems and methods used in many studies. Here, indicate whether each material, system or method listed is relevant to your study. If you are not sure if a list item applies to your research, read the appropriate section before selecting a response.

## Materials & experimental systems

| n/a | Involved in the study |
|---|---|
| ☐ | ☒ Antibodies |
| ☐ | ☒ Eukaryotic cell lines |
| ☒ | ☐ Palaeontology and archaeology |
| ☐ | ☒ Animals and other organisms |
| ☒ | ☐ Clinical data |
| ☒ | ☐ Dual use research of concern |
| ☒ | ☐ Plants |

## Methods

| n/a | Involved in the study |
|---|---|
| ☒ | ☐ ChIP-seq |
| ☒ | ☐ Flow cytometry |
| ☒ | ☐ MRI-based neuroimaging |

## Antibodies

| | |
|---|---|
| Antibodies used | rabbit Piezo1 (Novus, NBP1-78446); mouse S1P (Santa Cruz, CA sc-48356); rabbit KCNA1 (Alomone Labs, APC-161); rabbit KCNA2 (Alomone Labs, APC-010); rabbit LRRC8A (Alomone labs, AAC-001); mouse ZO1 (Invitrogen, 33-9100); rabbit ENaC antibodies SCNNA1 (Invitrogen, PA1-920A), SCNNB1(Invitrogen, PA5-28909), and SCNNG1(Invitrogen PA5-77797). Alexa Fluor 488, 568 and 647 goat anti–mouse and anti–rabbit IgG were used as secondary antibodies (Invitrogen). F-actin was stained using either conjugated 488 or 568 phalloidin (66μM) at 1:500 and DNA with 1μg/ml DAPI (Thermofisher).Primary rabbit anti-E-Cadherin antibody (24E10, Cell Signaling 3195) at 1:1000. |
| Validation | When available the blocking peptides were utilized to provide a negative control. Further matching findings and temporal spatial information from previous published studies and commercial website images were used to set precedent of antibodies binding success. https://www.novusbio.com/products/piezo1-antibody_nbp1-78446; https://www.scbt.com/p/edg-1-antibody-a-6?srsltid=AfmBOoof2AHM7OLfedK14dg0YC89JnzdivqHm1a-gJCOCk51NdO7UAEj; https://www.alomone.com/p/anti-kv1-1-extracellular/APC-161; https://www.alomone.com/p/anti-kv1-2/APC-010; https://www.alomone.com/p/anti-lrrc8a-extracellular-antibody/AAC-001; https://www.thermofisher.com/antibody/product/ZO-1-Antibody-clone-ZO1-1A12-Monoclonal/33-9100; https://www.thermofisher.com/antibody/product/alpha-ENaC-Antibody-Polyclonal/PA1-920A; https://www.thermofisher.com/antibody/product/SCNN1G-Antibody-Polyclonal/PA5-77797 https://www.cellsignal.com/products/primary-antibodies/e-cadherin-24e10-rabbit-mab/3195?srsltid=AfmBOooDupf8T_CfuYrDGQQbU5w7wI9iV5cgBMF6ZpCR4kEQKij-v2-- Gudipaty, S. A. & Rosenblatt, J. Epithelial cell extrusion: Pathways and pathologies. Semin Cell Dev Biol 67, 132-140 (2017). https://doi.org:10.1016/j.semcdb.2016.05.010 Iorio, J. et al. K(V)11.1 Potassium Channel and the Na(+)/H(+) Antiporter NHE1 Modulate Adhesion-Dependent Intracellular pH in Colorectal Cancer Cells. Front Pharmacol 11, 848 (2020). https://doi.org:10.3389/fphar.2020.00848 Serra, S. A. et al. LRRC8A-containing chloride channel is crucial for cell volume recovery and survival under hypertonic conditions. Proc Natl Acad Sci U S A 118 (2021). https://doi.org:10.1073/pnas.2025013118 Eskandari, N. et al. Molecular Activation of the Kv11.1 Channel Reprograms EMT in Colon Cancer by Inhibiting TGFbeta Signaling via Activation of Calcineurin. Cancers (Basel) 13 (2021). https://doi.org:10.3390/cancers13236025 Sinha, M. et al. Chloride channels in the lung: Challenges and perspectives for viral infections, pulmonary arterial hypertension, and cystic fibrosis. Pharmacol Ther 237, 108249 (2022). https://doi.org:10.1016/j.pharmthera.2022.108249 |

## Eukaryotic cell lines

Policy information about cell lines and Sex and Gender in Research

| | |
|---|---|
| Cell line source(s) | MDCK-II: European Collection of Authenticated Cell Cultures (ECACC) operated by Public Health England, catalogue number 00062107, lot 19G037. |
| Authentication | MDCKII authenticated before shipping before shipping. Cells were not authenticated after reception. |
| Mycoplasma contamination | Mycoplasma contamination was tested monthly in the cell line. All cell lines and passages were negative for mycoplasma. |
| Commonly misidentified lines (See ICLAC register) | no commonly misidentified cell lines were used in the study. |

## Animals and other research organisms

Policy information about studies involving animals; ARRIVE guidelines recommended for reporting animal research, and Sex and Gender in Research

| | |
|---|---|
| Laboratory animals | Ex vivo lung slices were obtained from mice (B6N.129S6(Cg) - Scgb1a1 <tm1 (cre/ERT) Blh>J) from 7 to 17 weeks of age. Housed following guidance of King's College London project license P68983265 in ambient temperature (roughly 20C) keeping in normal a normal circadian rhythm. |
| Wild animals | No wild animals were used in this study. |

| Reporting on sex | No experimental design was implemented to test sex differences. Both male and female lungs slices were used. |
|---|---|
| Field-collected samples | No field collected animals were used for this study. |
| Ethics oversight | All animals were housed under specific pathogen-free conditions and cared for in accordance with the UK Home Office Animals (Scientific Procedures) Act of 1986 and the guidelines set by the Institutional Committees on Animal Welfare. Animal experiments received approval from the Ethical Review Process Committee at King's College London and were conducted under a Home Office license in the UK. All tissues collected for experiments were taken and preformed post-mortem. Following all approved and provided guidance from King's College London  project license (P68983265) as per the Home Office. |

Note that full information on the approval of the study protocol must also be provided in the manuscript.

## Plants

| Seed stocks | *Report on the source of all seed stocks or other plant material used. If applicable, state the seed stock centre and catalogue number. If plant specimens were collected from the field, describe the collection location, date and sampling procedures.* |
|---|---|
| Novel plant genotypes | *Describe the methods by which all novel plant genotypes were produced. This includes those generated by transgenic approaches, gene editing, chemical/radiation-based mutagenesis and hybridization. For transgenic lines, describe the transformation method, the number of independent lines analyzed and the generation upon which experiments were performed. For gene-edited lines, describe the editor used, the endogenous sequence targeted for editing, the targeting guide RNA sequence (if applicable) and how the editor was applied.* |
| Authentication | *Describe any authentication procedures for each seed stock used or novel genotype generated. Describe any experiments used to assess the effect of a mutation and, where applicable, how potential secondary effects (e.g. second site T-DNA insertions, mosiacism, off-target gene editing) were examined.* |

