## [Peer Review File · Nature]

ENERGY DEFICIENCY SELECTS CROWDED LIVE EPITHELIAL CELLS FOR EXTRUSION

Corresponding Author: Professor Jody Rosenblatt

Version 0:

Reviewer comments:

Referee #1

(Remarks to the Author)

In this manuscript, Mitchell and coworkers followed up on their previous bahnbrechend work on the mechanisms of live cell extrusion in epithelial cells. By a combination of different imaging techniques, they delineate a pathway in which initial epithelial depolarization via ENaC activates voltage-gated Kv1.1, Kv1.2 and Swell1 channels, thus driving water egress and cell shrinkage which precedes cell extrusion (by a yet unknown mechanism that, however, involves Piezo1 and other K⁺ and Cl⁻ channels, as well as S1P/S1P2 signaling).

Overall, this is an interesting, novel and likely physiologically relevant mechanism that follows logically the previous work from this group. The main question, namely how individual cells are “selected” for extrusion, remains unanswered, and the proposed scheme based on neighboring cells with different vs. similar resting membrane potentials is largely based on speculation. There are also quite a number of loose ends that arise from the data but are not followed up; yet they may be relevant in the context of the present study.

Major comments:

1. The authors start off with “an important outstanding question”, namely “which cells within a crowded epithelial field are chosen for extrusion?” While the authors outline a novel and important pathway of cell extrusion, I do not feel that this central question is actually answered. The authors ultimately propose (Fig. 4K) that specific cells which are already more depolarized from the start will reach the threshold for Kv activation, and hence, shrinkage and extrusion earlier; yet, the assumption that these cells are already more depolarized is purely speculative as no data are shown (and would be technically difficult to generate) that specific cells that will later extrude are already pre-depolarized prior to crowding (as suggested in Fig. 4K). As such, while the scheme in Fig. 4K is intriguing, it’s first steps remain to be shown, and the central question “which cells within a crowded epithelial field are chosen for extrusion?” remains to be answered. In the abstract, this theory is further modified claiming that extrusion happens due to “insufficient energy to maintain cell membrane potential” which seems to hint more at a defect of the Na⁺/K⁺ ATPase as compared to the activation of ENaC later described in the paper.
2. The same holds true for the authors’ concept that in a monolayer of crowding cells with similar potential, cells will only shrink moderately and no extrusion will happen. To claim this, authors would have to compare absolute membrane potentials in each cell of the monolayer, which is presently (to my knowledge) not feasible.
3. As shown in Fig. 1F, cells tend to relax after shrinkage yet prior to extrusion. Is this “relaxation” response essential for the subsequent extrusion, and how is it mediated?
4. Fig. 1H shows that shrinkage alone only triggers extrusion in the presence of crowding. If the main effect of crowding is ENaC activation to induce depolarization and then, shrinkage, why is crowding required as an additive factor to shrinkage?
5. Not all extruding cells shrink (apparently, 30% do not as reported on page 5). The authors show that cell shrinkage rarely occurs prior to apoptotic extrusion, but they do not address the question how many of the 30% of cells that extrude without shrinkage were indeed apoptotic? Is there a relevant number of non-apoptotic LCE in the absence of shrinkage?
6. In Fig. 1I, the authors used inhibitors of SAC, ROCK or myosin II to show that actin-myosin contraction does not drive cell shrinkage. Notably, however, each of the applied substances in fact increased shrinkage in the lightning assay. Can the authors explain this counterintuitive finding?
7. Hypertonicity experiments – the timeline of these experiments is not entirely clear. It seems that cells were exposed to 10 min isotonicity, then 10 min hypertonicity, then back to isotonicity during which extrusion was monitored for 2 h. The

respective legend for Fig. 2B is, however, somehow confusing (“in normal medium or 100’ following 10 minute hypertonic shock”). More importantly, it seems that Suppl. Video 5 shows the interval after return to isotonicity; yet, it is then unclear why cells only start to shrink after return to hypertonicity, rather than during hypertonicity?

8. From page 2C, the authors conclude that “Only cells that shrank $20\pm 3\%$ as measured by cytoplasmic GFP volume extrude, whereas those shrinking $11\pm 2.5\%$ do not (Fig. C2C).” This is true for the mean values, yet not for individual cells as there is apparently considerable overlap between cells experiencing between 10-15% shrinkage some of which then undergo LES, while others do not. Hence, other (additional) mechanisms seem to be at play. What is the predictive value of the proposed cutoff of 17% volume change for cell extrusion (e.g. in a ROC curve)

9. Please clarify figures: Fig. 4F is (according to figure legend) supposed to show pharmacological inhibition of Kv1.1/Kv1.2 or ENaC with 4AP or amiloride, yet the actual figure states “siRNA” for the respective channels. Also, “maintained in monolayer” seems to apply to all conditions except extruding control cells, so does that mean that extruding cells in Kv1.1/Kv1.2 or ENaC inhibited/silenced conditions were not measured?

Fig. 5C and Fig. 5A seem to be switched. More importantly, Fig. 5A is supposed to show that 4AP does not prevent membrane depolarization, yet the effect on DiBAC3 fluorescence is almost identical to that of amiloride.

Minor comments:

1. Suppl. Fig. 1B: What does “similarly show dry mass before cell extrusion” in the figure legend refer to (similar to what?)

2. Fig. 1E: y-axis should be “junctional brightness” rather than “cell shrinkage”

3. Fig. 1F,G: In both cases, data were normalized to baseline, yet in F baseline values still vary but not in G. Please specify normalization for each.

4. Page 9 refers to Suppl. Fig 2C which does not seem to exist.

5. Suppl Fig. 3B: The authors report a $p=0.0024$ for the difference between DMSO and TRAM34 based on a two-way ANOVA with Dunnett’s multiple comparisons test. This seems hard to believe given the considerable overlap of data and the low n-number in the TRAM 34 group.

6. Fig. 3F: I guess there is an “siRNA” missing under the horizontal bar on the lower right. In Fig. 3E&F, was Swell1 siRNA or DCPIB treatment, respectively, always in the presence of zVAD-fmk?

7. The authors state (page 11) that “despite epithelial cells being connected with gap junctions, they are not fully electrically coupled” – do we know whether MDCKII cells in these culture experiments form functional gap junctions as epithelial cells would in vivo?

8. Suppl. Video 7: The video is supposed to highlight crowding and extruding cells by red and white arrows, which however were absent in my video.

Referee #2

(Remarks to the Author)

Maintaining proper epithelial cell number and morphology is quite important for development, epithelial barrier formation and prevention of tumorigenesis. The system is achieved by coordination of various cellular functions, including proliferation, cell death and cell competition.

The authors have been studying on epithelial cell barrier homeostasis mediated by cell extrusion and division. They originally found a stretch-activated channel (Piezo1) plays important roles in cell extrusion and cell division under epithelial homeostasis. The downstream signaling pathway of Piezo1 to induce cell extrusion have already suggested in the previous paper by the authors (S1P to Rho-mediated actomyosin contraction). However, it is not clear which cells are chosen for extrusion within a crowded epithelial field.

In present manuscript, they found transient cell shrinkage (homeostatic early shrinkage: HES) is an early hallmark before the live cell extrusion (LCE) by careful observation. Authors suggest that Kv1.1/Kv1.2 and SWELL1 function in water egression from cells to induce HES under ENaC-mediated depolarization, which is upstream of Piezo1 activation. The result showing artificial cell volume decreases by hypertonic solution increased cell extrusion (osmotic-induced cell extrusion: OICE) implies the linkage between cell volume regulation and cell extrusion. Based on the involvement of several channels (Kv1.1, Kv1.2, Sewll1 and ENaC subunits) in HES and extrusion, authors emphasize the importance of early depolarization and hydrodynamics in epithelial morphogenesis balanced by cell extrusion. The model linking membrane potential and hydrodynamic regulation to cell extrusion seems to be attractive and partially demonstrated. However, some technical limitations of the experimental methods (the lightning assay and measurement using DiBAC4(3)) and speculative discussion left concerns on the validity of the model. Thus, further experiments and proper interpretation of the results are required to strengthen the validity of authors’ model.

Major points

1. In Fig. 2A, why did transient hypotonic treatment fail to suppress extrusion? Prior hypotonic treatment may prevent cell shrinkage required for extrusion. How do authors interpret the result?

2. In Fig. 3C and E, the rationale is unclear why inhibitors and siRNA against the channels could prevent “cell shrinkage” (measured by the lightning assay) under hypertonic condition? Cell shrinkage by hypertonic stress is an immediate and simple physical phenomenon based on semipermeable property of the cell membrane, which does not require any additional ion export. Some explanation is required how authors interpret the area change in the lightning assay, especially in the case applied for whole regions of cell crowding,

3. Measurement of membrane potential by DiBAC3(4) is affected by loading amount of the dye. According to the METHODS & MATERIALS, authors normalized the intensity by 10 baseline frames of each cell. But in this manuscript, authors suggest baseline membrane potential may be different between extruded cells and surrounding cells. Furthermore, cell volume change may affect the intensity (the smaller cell volume makes brighter). These technical limitations of DiBAC3(4) prevents the accurate comparison between cells in Fig. 4. Other methods to measure membrane potential should be adopted to confirm the involvement of membrane potential.

4. How is ENaC channel activated in extruded cells in crowded monolayer? Or is it constitutively active? Is there any difference in ENaC activity between surrounding cells and cells to be extruded?
5. Though authors expect the Piezo1-S1P signaling is downstream of transient cell shrinkage, there is no evidence they are aligned in the same signaling pathway. It is possible that they may function in parallel under crowding-induced cell extrusion. Can siRNA or inhibitors against ENaC, Kv1.1/Kv1.2 or SWELL1 channels prevent Piezo1 activation and S1P production?
6. As authors discussed in the manuscript, it is strongly considerable that aquaporins are involved in HES. But there is no direct evidence on the significance of water transport. A conventional aquaporin inhibitor (HgCl₂) is suitable to investigate the involvement of water transport in HES (by the lightning assay) and cell extrusion.
7. In the field of cell competition, it is recently reported that Ca²⁺ waves and sparks (detected by Ca²⁺ probes) occurs prior to cell extrusion (Takeuchi et al., *Curr. Biol.*, 30, 670-681, 2020; Kuromiya et al., *Cell Rep.*, 40, 111078, 2022). It is interesting to investigate whether Ca²⁺ transient also be involved in the homeostatic extrusion studied in this manuscript. If it is the case, which step (ENaC-mediated depolarization or Piezo1 activation) does Ca²⁺ transient function?
8. The penetrance of the suggested model (early voltage and hydrodynamic regulation play roles in cell extrusion) should be examined in other epithelial cell models. As authors discussed that epithelial cells other than kidney may use different channels, it might be hard to identify channels involved in other cells. However, the hydrodynamic regulation by osmotic solution and aquaporin inhibition (see point 5) can be applicable to other cell models.

Minor points

There are plenty of miscitation of figures. Careful correction

1. Fig. 1H is difficult to understand. What does black dots represent? I think there should be plenty of “No shrink and No Extrusion” cells in the field. Is there any data representing uncrowded epithelial regions?
2. In page 5, line 4 and 5, authors describe, “we find cell shrinkage rarely occurs before apoptotic extrusion”. However, it is broadly known that cell shrinkage is a one feature of apoptotic cell death. It is better to precisely describe the time window when authors measure the lightning assay. I suppose it is earlier than execution of apoptosis.
3. Some figure citations in the text are incorrect, such as “Fig. C2C”, “Fig. A3A” and “Supplemental Fig. 3A&BB”.
4. In page 9, line 16, “Fig. 1D & E” might be “Fig. 1C & D”.
5. In page 9, line 24, Is “Suppl. Fig. 4B & C” correct? Sup. Fig. 4C is images of ENaC.
6. In page 11 and 12, some reference numbers are not superscript.
7. In Fig. 4E, there are two dashed vertical lines. Right one describes time of cell shrinkage. But what is the left one?
8. I could not find red and white arrows in Supplemental Video 7.
9. In page 13, line 5, “Supplementary Fig. 4D” is “Supplementary Fig. 4C”. And line 7, 4B & C should be 4C & D.
10. In page 13, line 6-7, “knock down of any of its subunit prevented both HES and OICE (Fig. 4G & H)” seems overstatement because cell shrinkage (HES) under isotonic solution is not suppressed by ENaC siRNAs (Fig. 4G).
11. In page 13, line 8-9, “in response to hypertonic challenge or during steady state turnover (Fig. 4C & F)”. Do Fig. 4C or F exhibit data under hypertonic condition? Their legend does not mention the hypertonic condition.
12. In page 13, line 16, regarding “and Supplementary Figs. 5A, B, & D”, it seems inappropriate to include Fig. S5A in this context.
13. In Supplementally Video 8, the buckling is not so clear. It is better to color the image depending on the Z-height like Fig. 4I.
14. Authors describes ENaC as a tension sensitive sodium channel in page 14, line 9. But in the case of this manuscript, cell crowding does not lead to stretch but compression. There are reports suggesting ENaC is activated both stretch and compression. Thus, it would be better to describe as “a mechano-sensitive channel” in consistent to other part of this manuscript.

Referee #3

(Remarks to the Author)

Summary of the key results:

The authors address the question of how particular cells within a crowded region are selected for extrusion. They first show that individual cells transiently shrink via water loss before they extrude and that inducing artificially cell shrinkage by increasing extracellular osmolarity is sufficient to induce cell extrusion. Then, they decipher the mechanism involved and show that the mechano-sensitive Epithelial Sodium Channel, ENaC, acts as the earliest crowd-sensing step, driving membrane depolarization, which in turn activate, the voltage-gated potassium channels Kv1.1 and Kv1.2 and the chloride channel SWELL1, upstream of Piezo1. Loss of any of these channels in crowded conditions causes epithelial buckling, highlighting an important role for voltage and water regulation in controlling epithelial shape as well as extrusion. The results are convincing and novel. They are of general interest, addressing how epithelia maintain their homeostasis through a detailed mechanism of cell extrusion dependent on membrane potential. However, the whole study relies on the analysis of cell extrusion in MDCK monolayers. How this applies to a physiological context? This would be important to address.

Fig1G: n=5 cells. This is not sufficient to conclude about a correlation between “lightning cell area” and cell volume.

I have also minor comments and suggestions that could make it easier for the reader.

Minor Comments:

- Fig1: How is determined the moment of extrusion? Are samples image in 3D to ensure that the extruding cell is out of the epithelium? This is an important point to clarify since the whole study relies on this.
- Fig1C: why using this method instead of measuring cell volume directly? What is the reason for this “lightning”? How to explain the difference with apoptotic cells presented in SupFig2?

- SupFig1: The authors adapted Quantitative Phase Imaging (QPI) in MDCK to analyze dry mass. Although the increase in cell mass is clear after extrusion, the increase in cell mass before cell division is not clear in panel C. I can not see clearly if this cell is dividing. An outline would help.
- In SupFig2A, the caspase marker disappears before the extrusion, which is quite surprising. Since caspase can be activated transiently in living cells, are you sure this cell is going to die?
- p5 "Transient cell volume loss could be due to myosin II contraction, which occurs before apoptotic extrusion" (1) is in contradiction with the previous conclusion that shrinkage occurs only before live extrusion. This conclusion is itself in contradiction with previous publication describing the loss of cell volume during apoptosis. Please clarify this point.
- p5 "HES occurs independently from myosin contractility" should be reformulated as "HES is not dependent on myosin contractility". Indeed, the shrinkage is not independent since it increases when Myosin II is inhibited, suggesting that Myosin II contractility prevents cell shrinkage.
- I would suggest to move Fig2D to Fig3 to be able to compare hypertonic induced extrusion from homeostatic extrusion.
- Fig3C: "C, Mean percent peak change in area surrounding cells". Please explain a bit more. How is defined the peak area in hypertonic conditions since no change in area is observed (Fig3B)? Here again, why using this essay as a proxy of cell volume. Measuring the area directly would be easier and more direct.
- Presentation is not always very straight forward. For example, Fig3D-E could be fused in two panels, one showing the effect of RNAis in cell shrinkage, the other on extrusion, in both conditions.
- Fig4 is very dense and could be split in 2, while Fig1 and Fig2A-C could fuse.

Referee #4

(Remarks to the Author)

In this study, Mitchell et al. propose that a transient osmotic cell shrinkage triggered by cell depolarization regulates cell extrusion, and thus epithelial homeostasis. Previously, the authors have shown that cell crowding triggers live cell extrusion in a piezo1-dependent manner, leading to homeostatic maintenance of epithelial cell density. However, how cells are selected for extrusion in dense areas is not fully understood. This study found that cell volume reduction at high cell density regions occurs before cell extrusion. They show that hypertonic treatment triggers cell extrusion in a manner dependent on Swell1 and Kv1.1 channels, and these channels are also required for homeostatic cell shrinkage and extrusion. Furthermore, cells show membrane depolarization through ENaC activity as they become denser, and the authors postulate the neighbor voltage differentials are critical determinants of the cell extrusion fate.

Their proposed model sounds interesting and might be particularly relevant in deepening our understanding of what determines cell extrusion fate. However, there are shortcomings, and many parts of the proposed model are speculative and inconclusive. For example, the involvement of ion channels in cell extrusion is investigated in multiple ways, but how these channels are regulated by cell crowding is not sufficiently analyzed. In addition, the authors have to be cautious about the effect of ion channel inhibition because it potentially affects transepithelial voltages, which is a critical regulator of epithelial tissue integrity. This said, the manuscript, at least in its current state, appears too preliminary to warrant publication in Nature.

Major comments

1. The abstract suggests that cell extrusion stems from insufficient energy to maintain cell membrane potential, but they have not tested anything about energy production. Could the authors check the energy state of the cells with, e.g. ATP or mitochondrial membrane potential sensors? Otherwise, they should refrain from describing the energy state in the abstract.
2. The authors used the 'lightning assay' to measure cell volume changes, but this is not convincing. The brightness in phase contrast images can also increase by the changes in cell shape. Thus, they should directly measure cell volume throughout the manuscript, as they have done in Figure 1G. Especially the authors claim that the induction of cell volume decrease by hypertonic treatment requires Kv1.1/2, Swell1, and ENaC activity (Figures 3C, 3E, and 4G), which is quite surprising to me and thus should be analyzed more carefully.
3. Could the author test how culturing cells in hypotonic media affects the homeostatic cell extrusions to verify the importance of cell volume regulation for the extrusions?
4. Whether neighbor voltage differentials can actually trigger live cell extrusion is not tested. The authors showed that cell depolarization correlates with cell extrusion, and the inhibition of ENaC or Kv1.1/2 can suppress the extrusion. Still, they do not indicate the sufficiency of the cell depolarization for extrusion. Therefore, the authors should examine how the induction of cell depolarization at a single cell level by, e.g. optogenetics affects cell volume and extrusion.
5. Whether and how cell crowding activates ENaC is entirely unclear. Although they assume that mechanical forces depending on cell crowding trigger the activation of ENaC and cell depolarization, this has not been experimentally tested. They should investigate how rapid changes in cell density by stretch or the release of stretch, as they have done in the previous papers (Eisenhoffer et al., Nature, 2012; Gudipaty et al., Nature 2017), affect ENaC activation (e.g. by sodium imaging with a sodium indicator with and without ENaC knock-down) and subsequent cell depolarization and extrusion. Also, they should analyze the dynamics of intracellular sodium concentration in homeostatic epithelial monolayers.
6. The authors must be more careful about the effect of ENaC and Kv1.1/2 knock-down on the monolayer shape (Figure 4I) because the knock-down potentially affects the establishment of transepithelial potential. As the authors also described in the manuscript, the alteration in the transepithelial potential can induce live-cell extrusions and 3D mounds (Saw et al., Nat.

Phys., 2022). Thus, Figure 4I might represent the results of the abrogation in the transepithelial potential. They should verify the importance of depolarization-dependent cell extrusion in this context. Moreover, the images with the xz section should be shown to indicate that some part of the monolayer is detached from the substrate and thus buckled, as illustrated in Figure 4I.

Minor comments

1. Regarding Supplementary Figure 1 and Video 1, they wrote, "QPI revealed that dry mass does not change before cell extrusion, unlike the clear mass increase before a cell divides." However, this is not conceivable from the presented figures and video. They should show quantification results of time-dependent changes in cell mass and multiple representative images before cell extrusion.
2. Regarding Figure 1I, they described that HES occurs independently from myosin contractility and Piezo1 activity, but this is not an appropriate expression. As they admitted in the manuscript, myosin and Piezo1 inhibitors increased the number of cells showing the brightness increase in phase images, which indicates the suppressive effect of myosin and Piezo1 activity in this context. They should rewrite the sentences.
3. In Figure 2C, statistic analysis is required to assess the difference in cell volume changes between cells extruded and maintained.
4. Although they claim that apoptotic cell extrusions account for ~20-30% of extruding cells at steady state, the presentation of Supplemental Figure 2 is insufficient to support this. They can re-analyze the data to show how much percentage of cell extrusion is apoptotic.
5. Figure 4A indicates that the crowded monolayer shows a heterogeneous membrane polarity state. Is there any correlation between local cell density and membrane potentials at a single-cell level?
6. In Supplementary Figure 4, Kv1.1 and Kv1.2 signals in separated and merged images look different, probably because they show other planes. The authors must show the same planes in each image. Also, they described that SWELL 1, Kv1.1, and Kv1.2 localize to the cell apex, but this is unclear from the represented images. To support this claim, they should show the images with higher magnification/resolution.
7. The concentration of amiloride should be described. It is not on the inhibitor table.

Version 1:

Reviewer comments:

Referee #1

(Remarks to the Author)

In the revised manuscript, the authors have added a substantial body of new data to address the key question how cells are selected for extrusion. They now show that cell crowding causes Na⁺ entry via ENaC that causes pronounced depolarizations in those cells that – due to insufficient ATP production – are unable to extrude Na⁺ effectively via the Na⁺/K⁺ATPase. The resulting depolarization then activates the voltage-gated K⁺ and Cl⁻ channels KV1.1, KV1.2 and Swell1, respectively, causing cells to shrink. In cells with sufficient shrinkage, the downstream Piezo1-S1P-S1P2-Rho pathway becomes activated to initiate cell extrusion. The proposed mechanism suggests a “darwinistic” concept for the physiological maintenance of epithelial barrier integrity, in that the “weakest” cells become extruded, while simultaneously preventing an indiscriminate activation of the Piezo1-S1P-S1P2-Rho extrusion pathway in “stronger” epithelial cells. Overall, the authors have done an excellent job to address my previous concerns. The manuscript has improved considerably and I only have a few minor concerns which you may please find below.

Comments:

1. Can the authors ascertain that it is really depolarisation-dependent activation of Piezo1 that ultimately drives extrusion, or could it also be the loss of ATP per se? I realize that the authors show that inhibition of Piezo (and its downstream signaling) block extrusion, but this could also be an indirect effect via Piezo-mediated ATP loss (by an unknown mechanism). This could be easily addressed with some control experiments showing that inhibition of Piezo does not prevent the loss of ATP and cell depolarisation during cell crowding.
2. In figure 4G, it seems that mitochondrial shape (based on ATP red staining) changes substantially between homeostatic and crowded cells, which may reflect a response to the greater energy challenge in cell crowding. Is this a consistent finding? If so, it should probably be commented on.
3. There is some inconsistency in the rebuttal letter as well as the manuscript with respect to the interpretation of the ENaC- and ATP-dependent changes in epithelial Na⁺ and the resulting membrane depolarisation. Specifically, at several instances in the rebuttal, the authors state that “decreasing ATP” or “blocking ATP synthase” causes “increased Na⁺ entry” – yet, according to the proposed concept, it is the lack of Na⁺ extrusion by the Na⁺/K⁺ATPase rather than an increased Na⁺ influx that causes the increase in cytosolic Na⁺ and the associated membrane depolarisation. Accordingly, in the revised manuscript in Fig. 4H, the y-axis should probably not read “average Na⁺ entry/1000 cells”, but rather “average increase in epithelial Na⁺ concentration/1000 cells”. Similarly, in line 246 the authors statement that “However, crowded cells challenged by ENaC activation with insufficient ATP levels to repolarize the membranes through the Na⁺/K⁺ ATPase will

experience a current that activates KV1.1 and KV1.2, causing cell shrinkage via water egress“ should probably be revised, as the activation of KV1.1 and KV1.2 is not the result of changes in Na⁺ current via ENaC, but due to changes in intracellular Na⁺ levels as a result of impaired Na⁺ extrusion.

Referee #2

(Remarks to the Author)

The authors have properly revised the manuscript, adding the measurements of ATP and intracellular Na concentration. These data addressed how extruding cells are selected and support the “speculation” in the previous manuscript. Ex vivo experiments using mouse bronchial epithelial cells suggested the relevance of the identified mechanism in some physiological context. The title change seems reasonable to clearly show the main point of the study. However, I still have concern in the interpretation of the lightning assay, though overall responses to comments and discussion are reasonable.

Minor point

Authors still suggest the lightning assay can be a proxy for general volume loss. I agree, in most part, the lightning correlates to volume loss in their experiments (in the extrusion context they studied). However, it is shown in many papers apoptosis is accompanied with cell shrinkage in the epithelial monolayer, though authors describe “cell shrinkage is rare before apoptotic extrusion” based on the lightning assay. Hypertonic treatment failed to induce acute cell shrinkage when several channels were inhibited. The acute shrinkage can be observed even for a simple liposome. As authors could not show the detailed rationale of the lightning under cell shrinkage (only speculate), I don't think all cell shrinkage should be detected by the lightning assay. I think it is better to carefully suggest the applicable subject for the lightning assay, otherwise authors should further investigate the generality of a one-to-one correspondence between the lightning and cell shrinkage in other contexts.

Referee #3

(Remarks to the Author)

Overall, the modifications made by the authors nicely improve the manuscript and the impossibility to perform some of the experiments asked well justified (such as aquaporin inhibition for example).

I still have a few minor comments that should be easy to address:

- In figure 4I: Osmolarity seems weirdly represented in the shrinking cell. I would have expected more electrolytes in the environment than in the extruding cell to explain the loss of water.
- Extrusions are not easy to spot in bronchiolar epithelial cells of mouse lung (Fig4E-F). What is the readout of extrusion here? How is extrusion quantified? The images could be improved to be more convincing.
- Fig3D: Na⁺ entry in future extruding cells should be quantified together with cell shrinkage to better highlight the sequence of events (Na⁺ entry first, shrinkage next). A representative example is insufficient.
- When talking about the impact of cell contractility on shrinkage, the author mention: “Thus, contractility appears to suppress cell shrinkage, presumably by stabilizing cell-cell junctions.” Since there could be many other ways for contractility to prevent shrinkage, the last part of the sentence should be removed.

Referee #4

(Remarks to the Author)

The authors have revised the manuscript in accordance with the suggestions provided by the various referees. They now present additional evidence indicating that the energy/ATP level of individual cells influences their likelihood of being extruded. While this represents an interesting extension to the proposed model of cell extrusion, it also raises several major concerns:

- The finding that uniformly lowering ATP levels results in increased cell extrusion is perhaps not entirely unexpected. A more compelling approach would be to manipulate ATP levels in a mosaic fashion and to test whether uniformly increasing ATP levels correspondingly reduces the rate of cell extrusion.
- The authors suggest—but do not directly demonstrate—that ATP is required for Na⁺/K⁺-ATPase activity to normalize membrane potential. How does reduced Na⁺/K⁺-ATPase activity affect cell extrusion? How sensitive is this enzyme's activity to fluctuations in intracellular ATP levels? Furthermore, could any potential reduction in cell extrusion due to elevated ATP levels be reversed by inhibiting Na⁺/K⁺-ATPase?
- Is cell extrusion governed by the absolute ATP level within a cell, or its relative level compared to neighboring cells?
- What is the relationship between the degree of Na⁺ influx/ membrane depolarization and intracellular ATP levels?

Given ATP's general role in tissue homeostasis and cell survival, its specific contribution to crowding-induced cell extrusion needs to be more clearly demonstrated.

Version 2:

Reviewer comments:

Referee #1

(Remarks to the Author)

For the revised manuscript, Mitchell and coworkers have performed additional control experiments to ensure that the

activation of Piezo is downstream of cell depolarization, and rephrased statements and figure annotations to clarify that the increase in cytosolic Na⁺ in extruded cells is attributable to the failure of Na⁺ extrusion via the Na⁺/K⁺ATPase rather than to increased Na⁺ entry.

All my remaining comments have been adequately addressed and I have no further concerns.

Referee #2

(Remarks to the Author)

The authors have properly revised the manuscript, carefully editing some sentences to resolve the previous my concern. I do not have any additional comment.

Referee #3

(Remarks to the Author)

The authors responded to my previous comments.

Referee #4

(Remarks to the Author)

The manuscript has been revised in accordance with the reviewers' suggestions. The current version shows clear improvement over previous iterations and now presents more compelling experimental evidence in support of the main claims.

Referees' comments:

Referee #1 (Remarks to the Author):

In this manuscript, Mitchell and coworkers followed up on their previous bahnbrechend work on the mechanisms of live cell extrusion in epithelial cells. By a combination of different imaging techniques, they delineate a pathway in which initial epithelial depolarization via ENaC activates voltage-gated Kv1.1, Kv1.2 and Swell1 channels, thus driving water egress and cell shrinkage which precedes cell extrusion (by a yet unknown mechanism that, however, involves Piezo1 and other K⁺ and Cl⁻ channels, as well as S1P/S1P2 signaling). Overall, this is an interesting, novel and likely physiologically relevant mechanism that follows logically the previous work from this group. The main question, namely how individual cells are “selected” for extrusion, remains unanswered, and the proposed scheme based on neighboring cells with different vs. similar resting membrane potentials is largely based on speculation. There are also quite a number of loose ends that arise from the data but are not followed up; yet they may be relevant in the context of the present study.

Thank you for the nice summary of our work. Your comments have helped us refocus the paper on the key signal that selects for extrusion. Our paper now shows that as epithelial cells become crowded, those with the lowest energy (ATP), are selected for extrusion mechanically by water-dependent volume loss. This mechanism highlights the importance for epithelia to maintain a strong barrier by ridding themselves of those with the weakest energy. Cells that cannot recover from mechanically activated membrane depolarization, electrically activate potassium (and chloride) channels that via water egress shrink cells sufficiently to activate the downstream Piezo1-S1P-S1P2-Rho pathway. Here, discovery of ENaC as the most upstream mechanosensitive channel acts to challenge the energy of each cell as they jostle within the epithelium. This mechanism not only selects against the weakest cells but may also act as a buffer against inadvertent Piezo1 activation from general buffeting of cells in a crowded region to trigger extrusion only in those shrinking at least 20%.

Major comments:

1. The authors start off with “an important outstanding question”, namely “which cells within a crowded epithelial field are chosen for extrusion?” While the authors outline a novel and important pathway of cell extrusion, I do not feel that this central question is actually answered. The authors ultimately propose (Fig. 4K) that specific cells which are already more depolarized from the start will reach the threshold for Kv activation, and hence, shrinkage and extrusion earlier; yet, the assumption that these cells are already more depolarized is purely speculative as no data are shown (and would be technically difficult to generate) that specific cells that will later extrude are already pre-depolarized prior to

crowding (as suggested in Fig. 4K). As such, while the scheme in Fig. 4K is intriguing, its first steps remain to be shown, and the central question “which cells within a crowded epithelial field are chosen for extrusion?” remains to be answered.

Thanks for your comment, which helped us identify why cells selected for extrusion have reduced membrane potential and helped us test all the critical steps of our model. Our paper now shows that as epithelial cells become crowded, the cells with lowest energy (ATP), are selected for extrusion mechanically by water-dependent volume loss. Here, we find that the Epithelial Sodium Channel, ENaC, acts as the most upstream mechanosensitive channel, challenging each cell as they jostle for position within the crowded epithelium by transiently depolarizing them. Cells with sufficient ATP can rectify sodium ingress with the Na⁺/K⁺ ATPase, which uses ~25% of cellular ATP. However, cells that lack enough ATP to counter depolarization, activate the voltage-gated potassium and chloride channels, KV1.1/KV1.2/Swell1, causing cells to shrink. Cells that shrink approximately 20% of their volume, then activate the downstream Piezo1-S1P-S1P₂-Rho pathway to elicit extrusion. This interesting pathway not only acts to evict the weakest cells but also prevents indiscriminately activating Piezo1-dependent extrusion of any crowded cell within the monolayer.

We now add results that support the following model:

1. Crowding causes Na⁺ entry into cells in an ENaC-dependent manner (Fig. 3D, G, &H).
2. Cells with reduced ATP accumulate Na⁺ and depolarise (Fig. 4A&B), which both crowding and experimentally reducing ATP increase (Fig. 4 G&H).
3. Depolarisation requires ENaC but not KV1.1 or KV1.2, indicating that ENaC produces the current upon crowding that activates potassium channels (Figure 3A-C).
4. Finally, cell shrinkage by water loss requires ENaC, KV1.1, KV1.2, and Swell1 channels, as knocking down or inhibiting any blocks both homeostatic and hypertonic-induced cell shrinkage and extrusion (Fig. 2 B&C and E&F & Supplemental Fig. 3H). Optogenetic K⁺ activation alone is not sufficient to activate shrinkage and extrusion; it is instead a readout of poor energy, as cells with sufficient energy can correct their K⁺ levels.
5. Cell shrinkage is sufficient to induce extrusion, as hypertonic treatment causes high rates of extrusion in both MDCK monolayers as well as bronchiolar epithelia from mouse ex vivo lung slices (Fig. 1M-N).

In our paper, we have tried to clarify these different steps with a schematic in each figure, as we gradually work upstream from cell shrinkage to loss of ATP, culminating in the entire pathway described in the first paragraph of the discussion. This indicates that as cells jostle about for space, the crowding activates ENaC, which causes transient membrane depolarization that can be rectified in cells with sufficient energy. Those without enough energy to repolarize their membranes experience a current that activates KV1.1/KV1.2 (and presumably Swell1) to cause shrinkage by water loss that activates the rest of the extrusion pathway.

In the abstract, this theory is further modified claiming that extrusion happens due to “insufficient energy to maintain cell membrane potential” which seems to hint more at a defect of the Na⁺/K⁺ ATPase as compared to the activation of ENaC later described in the paper.

We now report using two ATP reporters: ATP Red, a dye that measures mitochondrial ATP and Queen37, a genetically encoded quantitative ATP sensor, and find the same result with both. We now provide multiple lines of evidence that upon crowding, sodium enters cells but is retained only in cells with low ATP, causing the decrease in membrane potential that we identified in our first submission. Since we felt that the low energy aspect was the key selector, we decided to change the title to reflect this.

- We show that ATP decreases before membrane potential decreases (by DiBAC (3)4 staining) and cells shrink, using both a genetically encoded ATP probe, Queen37 (S. Fig. 5 A&B) and a mitochondrial ATP probe ATP Red (Figs. 4A&B).
- Decreasing ATP with oligomycin A in MDCK monolayers causes increased cell sodium entry (Fig. 4 G&H), shrinking (Fig. 4 C), and extrusion (Fig. 4D). Additionally, ATP is decreased in bronchiolar epithelial cells from ex vivo mouse lung slices before they extrude (Fig. 4F) and reducing ATP with Oligomycin A and oxamate increases extrusion rates in these slices (Fig. 4E).
- We also show that sodium enters a cell before it shrinks during homeostasis (Fig. 3D) and that blocking ATP synthase causes increased sodium cell entry in an ENaC-dependent manner (Fig. 4 G&H), where crowding increases this further (Fig. 3G&H).

We did try to briefly block Na⁺/K⁺ ATPase with ouabain, but all cells died immediately.

2. The same holds true for the authors' concept that in a monolayer of crowding cells with similar potential, cells will only shrink moderately and no extrusion will happen. To claim this, authors would have to compare absolute membrane potentials in each cell of the monolayer, which is presently (to my knowledge) not feasible.

As you state, measuring absolute membrane potential can be tricky, especially as epithelia are more likely to act in comparison to their neighbours rather than a given threshold setpoint. However, our new data shows CoroNa labelling of sodium ingress before cells extrude. Because we detect an increase in ENaC-dependent Na⁺ ingress following crowding or ATP reduction (which also increases extrusion rates (Figs. 3 & 4), we can follow this directly and find that while Na⁺ increases generally throughout the monolayer, it is only retained at higher levels in cells that go on to shrink and extrude. Further, in Fig. 1L, we have tried to change the graph readout so that it is clear to readers that following hypertonic challenge, only cells that shrink ~20% extrude, while those below this threshold are retained. We believe that these are important concepts to support the model, and we thank you for pointing them out.

3. As shown in Fig. 1F, cells tend to relax after shrinkage yet prior to extrusion. Is this “relaxation” response essential for the subsequent extrusion, and how is it mediated?

Shrinking cells with hypertonic solution without shifting back to isotonic medium causes rampant cell death within 30 minutes with no extrusion, suggesting that they do need to relax. Presumably, this is where the other regulatory volume ion channels, which we also identified as necessary for extrusion might play a role (see Fig. 2).

4. Fig. 1H shows that shrinkage alone only triggers extrusion in the presence of crowding. If the main effect of crowding is ENaC activation to induce depolarization and then, shrinkage, why is crowding required as an additive factor to shrinkage?

Our new findings that experimental crowding induces ENaC-dependent sodium entry into cells (Fig. 3G&H), which is required for both membrane depolarisation and cell shrinkage (Fig. 3A-F). Thus, we propose that ENaC acts at the first crowding tension sensor that tests which cells have low energy. Those that do not have sufficient ATP to repolarise their membranes, then activate the large (~20% volume) shrink that activates Piezo1. We think that this two-step mechano-regulation is an elegant way for ENaC to first select cells with least energy to extrude. Where cells that don't depolarize and dramatically shrink are mechanically buffered, dampening Piezo1 activation and commitment to extrusion from the minor cell jostling against one another. The downstream activation of Piezo1 warrants its own next paper, but we already know that it forms large cytoplasmic bodies only in crowded cell regions that degrade before extrusion, presumably by this significant shrink step.

5. Not all extruding cells shrink (apparently, 30% do not as reported on page 5). The authors show that cell shrinkage rarely occurs prior to apoptotic extrusion, but they do not address the question how many of the 30% of cells that extrude without shrinkage were indeed apoptotic? Is there a relevant number of non-apoptotic LCE in the absence of shrinkage?

We have tried to make this clearer by stating, “Using the lightning assay, we found that ~70% of cells shrink before live cell extrusion, whereas <0.03% of cells filmed (8 in 750) do without extruding.”

The bulk of cells, 70%, that undergo live cell extrusion shrink beforehand. We believe that indicates the major pathway for live cell extrusion, but it should be noted that other mechanisms can activate extrusion as well. For instance, cells with replicative damage¹ are likely to use a different pathway that depends on p53 activation. In another study we will submit soon, we find that starvation triggers a different set of defective cells through a different pathway to extrude, independent of crowding. Other pathways may also contribute. Here, our point is to investigate the reason for the most common pathway controls cell extrusion and death to maintain homeostatic levels.

To clarify this point, we also add a sentence to the end of the first paragraph: “Combining the lightning assay with a fluorescent reporter of the apoptotic marker cleaved caspase-3, we found that cell shrinkage is rare (~3%) before apoptotic

extrusion and could represent the 30% of cells that mysteriously do not shrink (Supplemental Fig 2A-B, Supplemental Video 3). Since most cells undergo HES which precedes live cell extrusion, this became of the focus of our investigation.”

6. In Fig. 1I, the authors used inhibitors of SAC, ROCK or myosin II to show that actin-myosin contraction does not drive cell shrinkage. Notably, however, each of the applied substances in fact increased shrinkage in the lightning assay. Can the authors explain this counterintuitive finding?

On line 103, we state, “To identify what causes HES, we investigated if canonical extrusion signals, like actomyosin contractility and Piezo1 activation, impact volume. Although others have found myosin activates a transient contraction before apoptotic extrusion^{11,12}, we found that inhibiting ROCK and myosin with Y-27632 and blebbistatin, respectively, instead increased the number of cells shrinking by ~23X, compared to untreated controls (Fig. 1H). Thus, contractility appears to suppress cell shrinkage, presumably by stabilizing cell-cell junctions.”

7. Hypertonicity experiments – the timeline of these experiments is not entirely clear. It seems that cells were exposed to 10 min isotonicity, then 10 min hypertonicity, then back to isotonicity during which extrusion was monitored for 2 h. The respective legend for Fig. 2B is, however, somehow confusing (“in normal medium or 100’ following 10 minute hypertonic shock”). More importantly, it seems that Suppl. Video 5 shows the interval after return to isotonicity; yet, it is then unclear why cells only start to shrink after return to hypertonicity, rather than during hypertonicity?

Thanks for pointing this out, we have fixed it. This delay was surprising to us as well. Presumably, isotonic recovery from the hypertonic shock causes a perturbation of water in all cells. Cells that would have enough energy could respond robustly and pump water back in, but those that struggle will have a difficult time responding and shrink and extrude. We suspect that the hypertonic treatment synchronizes and amplifies what occurs normally. It is for this reason that we felt it important to investigate the pathway under normal homeostatic conditions throughout the paper, rather than relying on this assay, despite how nicely it controls the timing of events.

8. From page 2C, the authors conclude that “Only cells that shrank $20\pm 3\%$ as measured by cytoplasmic GFP volume extrude, whereas those shrinking $11\pm 2.5\%$ do not (Fig. C2C).” This is true for the mean values, yet not for individual cells as there is apparently considerable overlap between cells experiencing between 10-15% shrinkage some of which then undergo LES, while others do not. Hence, other (additional) mechanisms seem to be at play. What is the predictive value of the proposed cutoff of 17% volume change for cell extrusion (e.g. in a ROC curve)

Thanks for pointing out that the way that we displayed this was confusing for our readers, as it emphasized movement long after the shrink event. Instead, we now plot only the percent shrink a cell experience while remaining or extruding in Fig. 1L and hope to make this point more clearly.

9. Please clarify figures: Fig. 4F is (according to figure legend) supposed to show pharmacological inhibition of Kv1.1/Kv1.2 or ENaC with 4AP or amiloride, yet the actual figure states “siRNA” for the respective channels. Also, “maintained in monolayer” seems to apply to all conditions except extruding control cells, so does that mean that extruding cells in Kv1.1/Kv1.2 or ENaC inhibited/silenced conditions were not measured?

Fig. 5C and Fig. 5A seem to be switched. More importantly, Fig. 5A is supposed to show that 4AP does not prevent membrane depolarization, yet the effect on DiBAC3 fluorescence is almost identical to that of amiloride.

Thanks for pointing out these labelling issues. We have tried to label more clearly to show that in controls, DiBAC(3)4 increases before a cell extrudes but that with:

- **4-AP blocks KV1.1 and KV1.2, but not membrane depolarization so DiBAC(3)4 still increases, accumulating in cells that depolarise but cannot extrude.**
- **amiloride blocks ENaC, thereby preventing membrane depolarization and increased DiBAC(3)4 fluorescence.**

Thus, ENaC controls depolarization whereas KV1.1 and KV1.2 do not.

Minor

comments:

1. Suppl. Fig. 1B: What does “similarly show dry mass before cell extrusion” in the figure legend refer to (similar to what?) **We have revised this supplemental figure and rephrased the text referring to it to: “Unlike proliferating cells that increase their dry masses before they divide, we found that dry mass does not change prior to cell extrusion (Supplemental Fig. 1A-C and video 1, representing 31 cells from 12 total cell islands), ruling out changes in cell mass as a selection factor for LCE.”**

2. Fig. 1E: y-axis should be “junctional brightness” rather than “cell shrinkage” **For measurements using the lightning assay, we refer to it now as HES, which we have now backed up with several methods to ensure it is a proxy for volume loss.**

3. Fig. 1F,G: In both cases, data were normalized to baseline, yet in F baseline values still vary but not in G. Please specify normalization for each. **The normalization of baseline cells was determined by dividing the cell area before shrink by the mean of 10 frames before the shrink cell area. As cells move it was important to show that this method was able to capture shrinkage not just background movement. This is what gives the variance seen in now Fig. 1G.**

4. Page 9 refers to Suppl. Fig 2C which does not seem to exist. **Thank you, this error has now been corrected.**

5. Suppl Fig. 3B: The authors report a $p=0.0024$ for the difference between DMSO and TRAM34 based on a two-way ANOVA with Dunnett’s multiple comparisons test. This seems hard to believe given the considerable overlap of data and the low n-number in the TRAM 34 group. **Thanks—now fixed.**

6. Fig. 3F: I guess there is an “siRNA” missing under the horizontal bar on the lower right. In Fig. 3E&F, was Swell1 siRNA or DCPIB treatment, respectively, always in the presence of zVAD-fmk? **Yes, in the supplemental figure 2C, we noticed that inhibiting or knocking down SWELL 1 alone increased apoptosis. Given that inhibiting apoptosis with ZVAD-FMK does not inhibit LCE, we combined treatment of KD or DCPIB with ZVAD-FMK to ensure we were quantifying live cell extrusion rates.**

7. The authors state (page 11) that “despite epithelial cells being connected with gap junctions, they are not fully electrically coupled” – do we know whether MDCKII cells in these culture experiments form functional gap junctions as epithelial cells would in vivo?

Yes, these MDCK monolayers should have functional gap junctions. We have revised this sentence to be more specific for individual cells, which helps describe our model better.

8. Suppl. Video 7: The video is supposed to highlight crowding and extruding cells by red and white arrows, which however were absent in my video.

In our revision, we no longer include the previous findings on how cells are buckled, as it cluttered and detracted from our main findings, which required more data in this revision.

Referee #2 (Remarks to the Author):

Maintaining proper epithelial cell number and morphology is quite important for development, epithelial barrier formation and prevention of tumorigenesis. The system is achieved by coordination of various cellular functions, including proliferation, cell death and cell competition.

The authors have been studying on epithelial cell barrier homeostasis mediated by cell extrusion and division. They originally found a stretch-activated channel (Piezo1) plays important roles in cell extrusion and cell division under epithelial homeostasis. The downstream signaling pathway of Piezo1 to induce cell extrusion have already suggested in the previous paper by the authors (S1P to Rho-mediated actomyosin contraction). However, it is not clear which cells are chosen for extrusion within a crowded epithelial field. In present manuscript, they found transient cell shrinkage (homeostatic early shrinkage: HES) is an early hallmark before the live cell extrusion (LCE) by careful observation. Authors suggest that Kv1.1/Kv1.2 and SWELL1 function in water egression from cells to induce HES under ENaC-mediated depolarization, which is upstream of Piezo1 activation. The result showing artificial cell volume decreases by hypertonic solution increased cell extrusion (osmotic-induced cell extrusion: OICE) implies the linkage between cell volume regulation and cell extrusion. Based on the involvement of several channels (Kv1.1, Kv1.2, Sewll1 and ENaC subunits) in HES and extrusion, authors emphasize the importance of early depolarization and hydrodynamics in epithelial morphogenesis balanced by cell extrusion. The model linking

membrane potential and hydrodynamic regulation to cell extrusion seems to be attractive and partially demonstrated. However, some technical limitations of the experimental methods (the lighting assay and measurement using DiBAC4(3)) and speculative discussion left concerns on the validity of the model. Thus, further experiments and proper interpretation of the results are required to strengthen the validity of authors' model.

Many thanks for this concise analysis and your comments, which we used to refine and better prove our model. Our new findings that experimental crowding induces ENaC-dependent sodium entry into cells (Fig. 3G&H), which is required for both membrane depolarisation and cell shrinkage (Fig. 3A-F). Thus, we propose that ENaC acts at the first crowding tension sensor that tests which cells have low energy. Those that do not have sufficient ATP to repolarise their membranes, then activate the large (~20% volume) shrink that activates Piezo1. We think that this two-step mechano-regulation is an elegant way for ENaC to first select cells with least energy to extrude. Where cells that don't depolarize and dramatically shrink are mechanically buffered, dampening Piezo1 activation and commitment to extrusion from the minor cell jostles against one another.

Major points

1. In Fig. 2A, why did transient hypotonic treatment fail to suppress extrusion? Prior hypotonic treatment may prevent cell shrinkage required for extrusion. How do authors interpret the result?

This is an interesting point you've raised, and we are not clear on why this is the case. The levels are somewhat but not significantly lower than remaining in isotonic (control) medium. It may be that once cells go from hypotonic to isotonic some still experience the same crowding forces. If left in hypotonic solution, we expect that they would not extrude, but as seen with the hypertonic treatment, cells are quite good at equilibrating their salts. We think that this is why hypertonic treatment works, as it synchronises cells to re-equilibrate their sodium and potassium, so those that don't have enough ATP are eliminated more rapidly. The main point we make in now Fig. 1J and 1L is that the hypertonic solution triggers some cells to shrink within 15' and then extrude, whereas the hypotonic solution did not.

2. In Fig. 3C and E, the rationale is unclear why inhibitors and siRNA against the channels could prevent "cell shrinkage" (measured by the lightning assay) under hypertonic condition? Cell shrinkage by hypertonic stress is an immediate and simple physical phenomenon based on semipermeable property of the cell membrane, which does not require any additional ion export. Some explanation is required how authors interpret the area change in the lightning assay, especially in the case applied for whole regions of cell crowding,

We were also surprised by this, as it has long been implied throughout the literature that water exchange does not require ion channels, yet we failed to find *any actual experiments* showing hypertonic stress being independent of ion channels. Our data clearly show here that they are needed for water egress. We have also now included

calcein dye, which is quenched when concentrated in cells with reduced volume, as another method to measure cellular volume and confirmed that water regulation depends on KV1.1, KV1.2, Swell1, and ENaC channels. We have now included this data in S. Fig. 3H. Our best guess is that if water leaves the cell through hypertonic treatment, the high ion content would trigger ion channel activation. If that doesn't happen, water might enter the cell again using another channel to safeguard the cell.

3. Measurement of membrane potential by DiBAC3(4) is affected by loading amount of the dye. According to the METHODS & MATERIALS, authors normalized the intensity by 10 baseline frames of each cell. But in this manuscript, authors suggest baseline membrane potential may be different between extruded cells and surrounding cells. Furthermore, cell volume change may affect the intensity (the smaller cell volume makes brighter). These technical limitations of DiBAC3(4) prevents the accurate comparison between cells in Fig. 4. Other methods to measure membrane potential should be adopted to confirm the involvement of membrane potential.

- Based on your comment, we tried using the genetically-encoded voltage indicators, Voltron and Aahn, but neither worked in our monolayers. Voltron works well in neurons and there is good data for Aahn in single non-neuronal cells, however, we have noted that many other genetically encoded probes that behave predictably in single cells completely re-localize or fail to work in epithelial monolayers, as they do not seem to be localizing/reading plasma membrane potential. The reasons between these discrepancies may be important and underexplored, since few people have done studies in mature, crowded epithelia. However, this differential staining (not at the plasma membrane) convinced us that they were not appropriate for *measuring plasma membrane potential* in epithelial monolayers. However, addition of the sodium probe, CoroNa clearly indicates plasma membrane depolarization, as sodium enters a cell before shrinkage, just like DiBAC (4)3 (Fig. 3D, and with crowding Fig. 3 G&H and Fig. 4G&H). We also use the sodium probe, CoroNa, to show that sodium influx occurs with crowding, ATP loss, and requires ENaC (Fig. 3H & I and Fig. 4H&I)).

We agree that many of the vital dyes for cells and organelles can be tricky to interpret, as many are sensitive to membrane potential or pH (mitotracker, lysotracker etc.) and some can alter what they read out over time. Yet, we feel that as we are measuring *differential* fluorescence between cells over the short time period before it shrinks, we can obtain meaningful data about relative potential between cells over time. The fact that we see DiBAC3(4) increase before we note any cell shrinkage suggests that its increased fluorescence is not dependent *only on* cell shrinkage. Further, as ATP red and the genetically encoded ATP sensor Queen 37 and calcein all decrease their fluorescence (while DiBAC3(4) increases) and remain decreased with shrink (Fig. 4A&B and S. Fig. 5A&B) also suggests that DiBAC3(4) fluorescence does not only reflect cell volume decrease. Finally, blocking the membrane potential decrease with the ENaC inhibitor amiloride also blocks shrinkage and DiBAC3(4) fluorescence,

whereas blocking the KV1.1 and 1.2 channels with 4-AP blocks cell shrinkage without preventing membrane depolarization and DiBAC3(4) fluorescence (See Fig. 3A-C). In fact, you can see it accruing even though cell shrinkage and extrusion is prevented. Thus, DiBAC3(4) is not intrinsically linked to cell shrinkage.

4. How is ENaC channel activated in extruded cells in crowded monolayer? Or is it constitutively active? Is there any difference in ENaC activity between surrounding cells and cells to be extruded?

This is an excellent point, which prompted us to use the sodium dye Corona AM, which we find increases in cells before they shrink and extrude (Fig. 3 D). Importantly, following experimentally crowding, while sodium increases slightly in all cells, it accumulates only in the cells with low ATP (Figs. 3 G&H and 4G&H). Blocking ENaC with amiloride prevents this crowding-dependent sodium increase. Importantly, reducing ATP levels with oligomycin A on its own increase sodium ingress, which is enhanced further with crowding.

Our interpretation is that crowding activates ENaC in all cells. Those that do not have sufficient energy to rectify it with the Na/K ATPase will maintain high sodium, activating the potassium and chloride channels to shrink. We could not even transiently block the Na/K ATPase with ouabain without immediately destroying the monolayer, strengthening its importance in cell viability.

5. Though authors expect the Piezo1-S1P signaling is downstream of transient cell shrinkage, there is no evidence they are aligned in the same signaling pathway. It is possible that they may function in parallel under crowding-induced cell extrusion. Can siRNA or inhibitors against ENaC, Kv1.1/Kv1.2 or SWELL1 channels prevent Piezo1 activation and S1P production?

The pathway that we describe in this paper controls the crowding-induced live cell extrusion that we have previously reported in (Eisenhoffer 2012), which controls most cell extrusions, which are live and crowding dependent. This is in line with our finding that shrinkage during homeostatic conditions only occurs in crowded regions as does extrusion and both pathways control ~70% of all extrusions occurring in the monolayer.

Furthermore, we inhibited S1P production with sphingosine kinases 1 and 2 inhibitors, Piezo1 with GSMTx4, and myosin with Y-27632 and blebbistatin (Fig.1 I and SFig.3 A&C) and found that shrinkage still occurs in this system and that they only block extrusion if present after the hypertonic treatment, placing shrinkage upstream all these extrusion signals. We now try to highlight this in our figures by adding a schematic of the pathway as we go upstream in each figure. What happens to Piezo1 upon shrinkage before a cell extrudes is complex and interesting but will deserve its own thorough study that will be the subject of a future paper. However, the graph above indicates that inhibiting ENaC (amiloride) or SWELL1 (DCPIB) significantly lowers the amount of S1P.

[REDACTED]

These changes may be subtle because all these inhibitors block extrusion so we cannot predict in which cells S1P would have been upregulated, but suggest that they act upstream of its production.

6. As authors discussed in the manuscript, it is strongly considerable that aquaporins are involved in HES. But there is no direct evidence on the significance of water transport. A conventional aquaporin inhibitor (HgCl₂) is suitable to investigate the involvement of water transport in HES (by the lighting assay) and cell extrusion.

Due to tight regulations of mercury, we were unable to obtain HgCl₂, despite numerous approaches. However, we have previously knocked down aquaporin 2 and inhibited aquaporin 3, which are most highly expressed in epithelia, and did not detect a noticeable difference. We do not feel that this data is strong enough to report on, as knockdown could enhance expression of other aquaporins, we may need to knockdown multiple in combination or we might be missing a critical one. Thus, we prefer to just mention it as a possibility, as it detracts from the main story.

7. In the field of cell competition, it is recently reported that Ca²⁺ waves and sparks (detected by Ca²⁺ probes) occurs prior to cell extrusion (Takeuchi et al., Curr. Biol., 30, 670-681, 2020; Kuromiya et al., Cell Rep., 40, 111078, 2022). It is interesting to investigate whether Ca²⁺ transient also be involved in the homeostatic extrusion studied in this manuscript. If it is the case, which step (ENaC-mediated depolarization or Piezo1 activation) does Ca²⁺ transient function?

Initially, we did extensive calcium imaging, expecting to see a spike before extrusion but instead found a dip! We now include it in supplemental Fig. 1D & E, along with our arduous journey into dry mass, which sadly also provided no insight.

8. The penetrance of the suggested model (early voltage and hydrodynamic regulation play roles in cell extrusion) should be examined in other epithelial cell models. As authors discussed that epithelial cells other than kidney may use different channels, it might be hard to identify channels involved in other cells. However, the hydrodynamic regulation by osmotic solution and aquaporin inhibition (see point 5) can be applicable to other cell models.

This is an excellent point and to address this along with reviewers' comments, we decided to confirm in tissue, rather than another cell line. Using bronchial epithelia from mouse ex vivo lung slices, we found that hypertonic treatment also induces a wave of epithelial cell extrusion (Fig. 1 M&N). Additionally, ATP reduction precedes extrusion in these bronchial epithelia and reducing ATP with oligomycin A and oxamate increases extrusion rates (Fig 4E & F).

Minor points

There are plenty of miscitation of figures. Careful correction **Thanks. We have added many new experiments and have tried to prevent this again.**

1. Fig. 1H is difficult to understand. What does black dots represent? I think there should be

plenty of “No shrink and No Extrusion” cells in the field. Is there any data representing uncrowded epithelial regions?

Thanks for this comment. We realize that it is confusing and that the label should have read shrinkage v. Density. We have now omitted the ‘no shrink and no extrusion’ as we realize it did not add much and just confused our readers.

2. In page 5, line 4 and 5, authors describe, “we find cell shrinkage rarely occurs before apoptotic extrusion”. However, it is broadly known that cell shrinkage is a one feature of apoptotic cell death. It is better to precisely describe the time window when authors measure the lightning assay. I suppose it is earlier than execution of apoptosis.

We ourselves have found that 4-AP, which blocks KV1.1 and KV1.2, blocks UV254-induced apoptosis and extrusion and this typically is required within the first minute of UV treatment, so very early on (Rosenblatt, 2001). Interestingly, following UV treatment of the monolayer all the cells seem to shrink apart from each other, much like the hypertonic treatments, suggesting that apoptotic cell extrusion through this pathway may use some similar ion channel regulation. However, here, we are following homeostatic apoptotic extrusion and we did not detect cell shrinkage occurring beforehand—at any time. Therefore, it suggests that there is a lot we do not really understand about these channels and all the different pathways to apoptosis.

3. Some figure citations in the text are incorrect, such as “Fig. C2C”, “Fig. A3A” and “Supplemental Fig. 3A&BB”.

Thanks. We have checked these and the new additions.

4. In page 9, line 16, “Fig. 1D & E” might be “Fig. 1C & D”.

Thanks. We have checked these and the new additions.

5. In page 9, line 24, Is “Suppl. Fig. 4B & C” correct? Sup. Fig. 4C is images of ENaC.

Thanks. We have checked these and the new additions.

6. In page 11 and 12, some reference numbers are not superscript.

Thanks. We have checked these and the new additions.

7. In Fig. 4E, there are two dashed vertical lines. Right one describes time of cell shrinkage. But what is the left one?

Thanks for noticing this did not get labelled. The right dotted line represents onset of depolarization. I have labelled as ‘depolarization starts’ now found as Fig.3B.

8. I could not find red and white arrows in Supplemental Video 7.

As the focus of the paper has changed this supplemental video has been removed.

9. In page 13, line 5, “Supplementary Fig. 4D” is “Supplementary Fig. 4C”. And line 7, 4B & C should be 4C &D.

Thanks. We have checked these and the new additions.

10. In page 13, line 6-7, “knock down of any of its subunit prevented both HES and OICE (Fig. 4G & H)” seems overstatement because cell shrinkage (HES) under isotonic solution is not suppressed by ENaC siRNAs (Fig. 4G).

These figures have been revised. The isotonic condition was only showing that the adapted lightning assay taken from a collective area of cells was not changed by drug and or siRNA treatment. The figure revision now just includes the OICE and HES graphs now to avoid any confusion and to demonstrate that blocking ENaC inhibits both HES (Fig. 3E) and OICE (S Fig. 4 C&E).

11. In page 13, line 8-9, “in response to hypertonic challenge or during steady state turnover (Fig. 4C & F)”. Do Fig. 4C or F exhibit data under hypertonic condition? Their legend does not mention the hypertonic condition.

In previous Fig. 4C and F, now Fig. 3A and C respectively, is tested under steady state turn over. The figure legends now include that this in homeostatic conditions.

12. In page 13, line 16, regarding “and Supplementary Figs. 5A, B, &D”, it seems

inappropriate to include Fig. S5A in this context.

Thanks for pointing out. In the cases where hypertonic solution were used to induce OICE. The lightning assay was adapted to capture an area not just around an extruding cell so here the isotonic condition was used to show that drug and or SiRNA treatment did not illicit lightning changes in normal steady state cells. This has been revised to graph OICE (Fig. 2E, Fig. 3E S.Fig. 4C) and HES (Fig. 2E, S. Fig. 4F). Here is an example of from Figure 2E of OICE.

13. In Supplementally Video 8, the buckling is not so clear. It is better to color the image depending on the Z-height like Fig. 4I.

This is a great suggestion that we will use when writing the follow up paper that will include the buckling data. As mentioned previously, we have decided to omit this data to instead focus on the extruding cell pathway.

14. Authors describes ENaC as a tension sensitive sodium channel in page 14, line 9. But in the case of this manuscript, cell crowding does not lead to stretch but compression. There are reports suggesting ENaC is activated both stretch and compression. Thus, it would be better to describe as “a mechano-sensitive channel” in consistent to other part of this manuscript.

We will refer to it as a mechanosensitive channel but now we provide clear evidence in Figs. 3G&H and 4G&H that crowding activates it, as the sodium dye indicates clear sodium ingress under crowding.

Referee #3 (Remarks to the Author):

Summary of the key results:

The authors address the question of how particular cells within a crowded region are selected for extrusion. They first show that individual cells transiently shrink via water loss before they extrude and that inducing artificially cell shrinkage by increasing extracellular osmolarity is sufficient to induce cell extrusion. Then, they decipher the mechanism involved and show that the mechano-sensitive Epithelial Sodium Channel, ENaC, acts as the earliest crowd-sensing step, driving membrane depolarization, which in turn activates the voltage-gated potassium channels Kv1.1 and Kv1.2 and the chloride channel SWELL1, upstream of Piezo1. Loss of any of these channels in crowded conditions causes epithelial buckling, highlighting an important role for voltage and water regulation in controlling epithelial shape as well as extrusion.

The results are convincing and novel. They are of general interest, addressing how epithelia maintain their homeostasis through a detailed mechanism of cell extrusion dependent on membrane potential. However, the whole study relies on the analysis of cell extrusion in MDCK monolayers. How this applies to a physiological context? This would be important to address.

Many thanks for your concise summary of our findings. We agree that confirming our results in real tissue is an important point and so have tested the main finding in bronchial epithelia in ex vivo mouse lung slices. We found that hypertonic treatment also increases extrusion rates in bronchial epithelial cells (Fig. 1M&N) and that reducing ATP with oligomycin A and oxamate similarly increases extrusion in mouse ex vivo lung slices (Fig 4G). Additionally, as seen in MDCK cells, ATP Red fluorescence decreases in cells before extruding from mouse bronchial epithelia at steady state (Fig. 4E).

Fig1G: n=5 cells. This is not sufficient to conclude about a correlation between “lightning cell area” and cell volume.

We increased the number of cells analyzing volume with cytoplasmic GFP before they shrink and extrude (Fig 1C, n=15 for each case). We now also include experiments using calcein dye, which quenches fluorescence with increased solute and has been used to measure osmotic volume changes over a short time scale. Here, too, we see that calcein reduces within cells before extruding (Figure 1D) and reduces as expected under hypertonic conditions (S. Fig. 2E).

I have also minor comments and suggestions that could make it easier for the reader.
Minor Comments:

- Fig1: How is determined the moment of extrusion? Are samples image in 3D to ensure that the extruding cell is out of the epithelium? This is an important point to clarify since the whole study relies on this.

Hindsight in timelapse movies is 20/20. We typically work backwards from a cell that has extruded completely out the monolayer and then backtrack to the initiation point,

marked as the first frame where the cell rounds before it goes on to extrude. To clarify, we will state this in the results.

- Fig1C: why using this method instead of measuring cell volume directly? What is the reason for this “lightning”? How to explain the difference with apoptotic cells presented in SupFig2?

We developed the lightning assay because it was easy to capture in live phase movies without requiring dyes, allowing us to readily score high n’s. We feel confident that this method directly reflects 1) our 3D volume measurement using cytoplasmic GFP at high resolution, which is extremely laborious to measure, and 2) calcein-AM dye, which reduces with water loss (Supplemental Fig. 2E). The ‘lightning’ we measure occurs when the cell is shrinking, suggesting that cells are pulling away from each other at these borders, allowing more light to pass through by phase. There may be other explanations that we do not understand yet, but it does represent cell shrinkage. The purpose of Sup Fig 2 is to show that apoptotic cells do not undergo a shrink before extruding like live cells do, suggesting that apoptotic cell extrusion uses a different pathway.

- SupFig1: The authors adapted Quantitative Phase Imaging (QPI) in MDCK to analyze dry mass. Although the increase in cell mass is clear after extrusion, the increase in cell mass before cell division is not clear in panel C. I cannot see clearly if this cell is dividing. An outline would help.

We have revised this figure to frame the cells as they go through both extrusion and division, which we hope clarifies dry mass changes in each case. Thanks for your attention to this.

- In SupFig2A, the caspase marker disappears before the extrusion, which is quite surprising. Since caspase can be activated transiently in living cells, are you sure this cell is going to die?

We typically notice that the caspase dye dissipates as the apoptotic cell blebs later in the extrusion, presumably as these cells become slightly permeable. Importantly, at 52’ in SFig. 2A, the cell is still caspase positive at the time it commits to extruding.

- p5 “Transient cell volume loss could be due to myosin II contraction, which occurs before apoptotic extrusion” (1) is in contradiction with the previous conclusion that shrinkage occurs only before live extrusion. This conclusion is itself in contradiction with previous publication describing the loss of cell volume during apoptosis. Please clarify this point.

Thanks for flagging this, we think it is more accurate to say that in zebrafish cells contract in a pulsatile manner before extruding. We have changed the text to:
“Although others have found myosin activates a transient pulsatile contraction before apoptotic extrusion^{11,12}, we found that inhibiting ROCK and myosin with Y-27632 and blebbistatin, respectively, instead increased the number of cells shrinking by ~23X, compared to untreated controls (Fig. 1H). Thus, contractility appears to suppress cell shrinkage, presumably by stabilizing cell-cell junctions.”

- p5 “HES occurs independently from myosin contractility” should be reformulated as “HES is not dependent on myosin contractility”. Indeed, the shrinkage is not independent since it increases when Myosin II is inhibited, suggesting that Myosin II contractility prevents cell shrinkage.

Good point. We have revised, as state above.

- I would suggest to move Fig2D to Fig3 to be able to compare hypertonic induced extrusion from homeostatic extrusion.

We now edited all the figures and now this is seen in Fig. 1. We agree that it is nice to see the differences.

- Fig3C: “C, Mean percent peak change in area surrounding cells”. Please explain a bit more. How is defined the peak area in hypertonic conditions since no change in area is observed (Fig3B)? Here again, why using this assay as a proxy of cell volume. Measuring the area directly would be easier and more direct.

We now clarify how we measure this in the methods, by saying we measure the percentage of shrinkage by thresholding and measuring the area of phase-brightness around cells before and after hypertonic solution, since this can vary between individual cells. To provide support that our lightning assay serves as a good proxy for cell volume loss, we have: 1) added more direct volume measurements using mosaically labelled cytoplasmic GFP expressing cells, where we can measure the volumed through by confocal using 0.2 μ Z sections throughout the process (extremely time consuming), and 2) by using calcein AM, which fluoresces in cytoplasm but is quenched with higher solutes and, therefore can measure changes in volume over short time frames. We also used calcein Am to confirm that cell shrinkage requires the ion channels we identified (Supplemental Fig. 3H).

- Presentation is not always very straight forward. For example, Fig3D-E could be fused in two panels, one showing the effect of RNAis in cell shrinkage, the other on extrusion, in both conditions.

We initially presented it as you suggested, but several commented that it was too confusing, with two different Y axes and aspects measured. Additionally, we felt it was important and surprising that ion channels control water regulation, and felt it needed its own graph.

- Fig4 is very dense and could be split in 2, while Fig1 and Fig2A-C could fuse. **Thanks for this and other comments. Having added in many extra experiments, we needed to reorganize the paper. To try to make it flow better, we have added schematics to show how each finding fits into a growing pathway, culminating in low ATP being the most upstream signal to activate extrusion. Due to these additions and refocusing our paper on only extruding cells, we have now removed background membrane potential measurements and buckling. We felt that this detracted from the main message of the paper and likely deserves its own study.**

Referee #4 (Remarks to the Author):

In this study, Mitchell et al. propose that a transient osmotic cell shrinkage triggered by cell depolarization regulates cell extrusion, and thus epithelial homeostasis. Previously, the authors have shown that cell crowding triggers live cell extrusion in a piezo1-dependent manner, leading to homeostatic maintenance of epithelial cell density. However, how cells are selected for extrusion in dense areas is not fully understood. This study found that cell volume reduction at high cell density regions occurs before cell extrusion. They show that hypertonic treatment triggers cell extrusion in a manner dependent on Swell1 and Kv1.1 channels, and these channels are also required for homeostatic cell shrinkage and extrusion. Furthermore, cells show membrane depolarization through ENaC activity as they become denser, and the authors postulate the neighbor voltage differentials are critical determinants of the cell extrusion fate.

Their proposed model sounds interesting and might be particularly relevant in deepening our understanding of what determines cell extrusion fate. However, there are shortcomings, and many parts of the proposed model are speculative and inconclusive. For example, the involvement of ion channels in cell extrusion is investigated in multiple ways, but how these channels are regulated by cell crowding is not sufficiently analyzed. In addition, the authors have to be cautious about the effect of ion channel inhibition because it potentially affects transepithelial voltages, which is a critical regulator of epithelial tissue integrity. This said, the manuscript, at least in its current state, appears too preliminary to warrant publication in Nature.

Thank you for these astute comments, which helped us fill the holes in our model better. Our revised paper now shows that as epithelial cells become crowded, the cells with lowest energy (ATP), are selected for extrusion mechanically by water-dependent volume loss. This mechanism highlights the importance for epithelia to maintain a strong membrane potential, with cells that cannot maintain this potential electrically activating potassium and chloride channels that cause cells to shrink sufficiently to activate the downstream Piezo1-S1P-S1P2-Rho pathway that causes extrusion. Here, discovery of ENaC as the most upstream mechanosensitive channel acts to challenge the energy of each cell as they jostle within the epithelium. This system could not only select against the weakest cells but also buffer crowding activation on Piezo1 so that it only activates cells experiencing ~20% shrinkage, rather than indiscriminate background crowding forces.

Major comments

1. The abstract suggests that cell extrusion stems from insufficient energy to maintain cell membrane potential, but they have not tested anything about energy production. Could the authors check the energy state of the cells with, e.g. ATP or mitochondrial membrane potential sensors? Otherwise, they should refrain from describing the energy state in the abstract. **Thanks for this comment. This and others helped us better pinpoint the cause of a cell's**

lack of membrane potential. Based on these comments, we have revised the story to focus on the mechanism for how low energy (ATP) selects a cell in a crowded region for live extrusion. To do this rigorously, we have decided to remove our other findings on what the background cells were doing with respect to shrinkage, which we felt served as a distraction to our main point. To specifically test all the salient points of our model, we now add the following results:

We now add results that support the following model:

1. Crowding causes Na⁺ entry into cells in an ENaC-dependent manner (Fig. 3D, G, &H).
2. Cells with reduced ATP accumulate Na⁺ and depolarise (Fig. 4A&B), which both crowding and experimentally reducing ATP increase (Fig. 4 G&H).
3. Depolarisation requires ENaC but not KV1.1 or KV1.2, indicating that ENaC produces the current upon crowding that activates potassium channels (Figure 3A-C).
4. Finally, cell shrinkage by water loss requires ENaC, KV1.1, KV1.2, and Swell1 channels, as knocking down or inhibiting any blocks both homeostatic and hypertonic-induced cell shrinkage and extrusion (Fig. 2 B&C and E&F & Supplemental Fig. 3H).
5. Cell shrinkage is sufficient to induce extrusion, as hypertonic treatment causes high rates of extrusion in both MDCK monolayers as well as bronchiolar epithelia from mouse ex vivo lung slices (Fig. 1J-N).

In our paper, we have tried to clarify these different steps with a schematic in each figure as we gradually work upstream from cell shrinkage to loss of ATP, culminating in the entire pathway described in the first paragraph of the discussion. This indicates that as cells jostle about for space, the crowding activates ENaC, which causes transient membrane depolarization that can be rectified in cells with sufficient energy. Those without enough energy to repolarize their membranes experience a current that activates KV1.1/KV1.2 (and presumably Swell1) to cause shrinkage by water loss that activates the rest of the extrusion pathway.

2. The authors used the 'lightning assay' to measure cell volume changes, but this is not convincing. The brightness in phase contrast images can also increase by the changes in cell shape. Thus, they should directly measure cell volume throughout the manuscript, as they have done in Figure 1G. Especially the authors claim that the induction of cell volume decrease by hypertonic treatment requires Kv1.1/2, Swell1, and ENaC activity (Figures 3C, 3E, and 4G), which is quite surprising to me and thus should be analyzed more carefully.

To confirm that our volume measurements by cytoplasmic GFP validated the lightning assay we presented, we have: 1) increased the number of measurements using confocal imaging of mosaically expressed cytoplasmic GFP to measure the volume from 0.2 μ Z sections over time (Fig 1C, n=15 for each case), and 2) used calcein, a fluorescent dye that is quenched with higher solutes and can, therefore be used to measure volume over short timeframes. We now show that calcein fluorescence reduces to a similar extent compared to our cytoplasmic GFP measurements during the phase-bright shrinkage measured before homeostatic extrusion and with

hypertonic challenge (Fig. 1D and SFig. 2E & 3H and below left). While we increased the cell volume measurements using cytoplasmic GFP, they are extremely time consuming and preclude the ability to simultaneously track membrane potential, ATP, and sodium ingress. Therefore, calcein and cytoplasmic GFP volume measurements were critical for helping us rely on the lightning assay to follow shrinkage in conjunction with these other parameters.

Further, we also used calcein to validate that inhibiting ion channels Kv1.1, 1.2, Swell 1, and ENaC suppress water loss even when challenged with hypertonic treatment (Supplemental Figure 3H and below, right). The channel dependence on cell volume was surprising to us as well, as it is assumed throughout the literature that osmotic changes occur independently, however, our searches found no experiments to actually support these claims.

3. Could the author test how culturing cells in hypotonic media affects the homeostatic cell extrusions to verify the importance of cell volume regulation for the extrusions?

We did include up to 35% hypotonic as well and did not see a significant impact on extrusion. See Fig. 1, also below:

4. Whether neighbor voltage differentials can actually trigger live cell extrusion is not tested. The authors showed that cell depolarization correlates with cell extrusion, and

the inhibition of ENaC or Kv1.1/2 can suppress the extrusion. Still, they do not indicate the sufficiency of the cell depolarization for extrusion. Therefore, the authors should examine how the induction of cell depolarization at a single cell level by, e.g. optogenetics affects cell volume and extrusion.

We expressed Blink2 to optogenetically activate potassium channels in specific cells with blue light but found that it was not sufficient to trigger extrusion, which we mention in the text, *“To test if potassium channel activation were sufficient to trigger cell shrinkage and extrusion, we optogenetically activated potassium channels in specific cells within MDCKII monolayers expressing Blink2². However, we found that only 1 cell out of 29 activated with blue light went on to shrink and extrude. Thus, activating K⁺ egress was not sufficient to induce extrusion and suggested that another signal must also operate.”*

Our new findings that only cells with insufficient ATP maintain sodium, depolarise, and shrink likely answers why this happens. If cells have sufficient energy, they should be able to readily re-polarize potassium via the Na⁺/K⁺ ATPase so that they will not extrude. We imagine if we targeted cells that had low ATP, optogenetic activation of potassium loss would cause shrinkage and extrusion but this would also happen without optogenetic activation and may be why only 1 in 31 cells did extrude after optogenetically activating K⁺ channels.

We have added the sodium probe, CoroNa as another readout of plasma membrane depolarization, as sodium enters a cell before shrinking and extruding, just like DiBAC (4)3 (Fig. 3D, and with crowding Fig. 3 G&H and Fig. 4G&H). Moreover, sodium influx (measured by CoroNa) occurs with crowding, ATP loss, and requires ENaC (Fig. 3H & I and Fig. 4H&I)). With crowding, it goes up in all cells slightly but remains higher in cells with low ATP, which then go on to shrink and extrude.

5. Whether and how cell crowding activates ENaC is entirely unclear. Although they assume that mechanical forces depending on cell crowding trigger the activation of ENaC and cell depolarization, this has not been experimentally tested. They should investigate how rapid changes in cell density by stretch or the release of stretch, as they have done in the previous papers (Eisenhoffer et al., Nature, 2012; Gudipaty et al., Nature 2017), affect ENaC activation (e.g. by sodium imaging with a sodium indicator with and without ENaC knock-down) and subsequent cell depolarization and extrusion. Also, they should analyze the dynamics of intracellular sodium concentration in homeostatic epithelial monolayers. **These were excellent suggestions-thanks. We have now included these experiments in Fig. 3 & 4. Using the sodium dye CoroNa, we found that sodium clearly enters before shrinking and extruding during homeostasis, although, as this dye bleaches over time, we were limited in the numbers we could find during homeostasis. Thus, we employed experimental crowding to rapidly and directly measure the response and found that crowding caused an overall sodium increase, where only cells with low ATP retained the increased sodium (Figs. 3&4 G&H). Reducing ATP with Oligomycin A increased the sodium ingress and retention further (Fig. 4 G&H).**

6. The authors must be more careful about the effect of ENaC and Kv1.1/2 knock-down on the monolayer shape (Figure 4I) because the knock-down potentially affects the establishment of transepithelial potential.

As the authors also described in the manuscript, the alteration in the transepithelial potential can induce live-cell extrusions and 3D mounds (Saw et al., Nat. Phys., 2022). Thus, Figure 4I might represent the results of the abrogation in the transepithelial potential. They should verify the importance of depolarization-dependent cell extrusion in this context. Moreover, the images with the xz section should be shown to indicate that some part of the monolayer is detached from the substrate and thus buckled, as illustrated in Figure 4I.

We appreciate this insight, however, with the many new included experiments that greatly clarified our main focus on ion channels and water loss and ordering these events clearly, we decided to remove the data on background (non-extruding) cell depolarization and buckling, as we felt it detracted from the main story and made the figures to dense. We do believe that these studies will make an interesting follow-on story and are grateful for your comments. The buckles, however, reflected the same data in Fig. 3A, where disruption of KV1.1 or KV1.2 causes DiBAC to accumulate but disruption of ENaC did not in the buckles. In Fig. 3A you can even start to see the beginnings of cells buckling with 4-AP.

Minor comments

1. Regarding Supplementary Figure 1 and Video 1, they wrote, "QPI revealed that dry mass does not change before cell extrusion, unlike the clear mass increase before a cell divides." However, this is not conceivable from the presented figures and video. They should show quantification results of time-dependent changes in cell mass and multiple representative images before cell extrusion.

As requested, we have added quantification of the mass of both the extruding cell and the dividing cell to the revised SFig.1. We also included additional images of the extruding cell prior to the extrusion event. Starting on line 62, we state "To test this possibility, we adapted quantitative phase imaging (QPI) to analyze dry mass changes over time in Madin-Darby Canine Kidney II (MDCKII) epithelial monolayers^{8,9}. Unlike proliferating cells that increase their dry masses before they divide, we found that dry mass does not change prior to cell extrusion (Supplemental Fig. 1A-C and video 1, representing 31 cells from 12 total cell islands), ruling out changes in cell mass as a selection factor for LCE."

2. Regarding Figure 1I, they described that HES occurs independently from myosin contractility and Piezo1 activity, but this is not an appropriate expression. As they admitted in

the manuscript, myosin and Piezo1 inhibitors increased the number of cells showing the brightness increase in phase images, which indicates the suppressive effect of myosin and Piezo1 activity in this context. They should rewrite the sentences.

We agree and have revised the statement on line 103 as “Unexpectedly, myosin II contraction, required for pulsatile contractions just before apoptotic extrusion in zebrafish skin^{3,4} does not cause HES, as inhibiting it with Y-27632 or blebbistatin instead increased the number of cells shrinking by ~23X, compared to untreated controls (Fig. 1H). Thus, contractile myosin suppresses cell shrinkage, presumably by stabilizing cell-cell junctions.”

3. In Figure 2C, statistic analysis is required to assess the difference in cell volume changes between cells extruded and maintained.

To confirm that our volume measurements by cytoplasmic GFP validated the lightning assay we presented, we have: 1) increased the number of measurements using confocal imaging of mosaically expressed cytoplasmic GFP to measure the volume from 0.2μ Z sections over time (extremely time consuming) (Fig 1C, n=15 for each case), and 2) used calcein, a fluorescent dye that is quenched with higher solutes and can, therefore be used to measure volume over short timeframes. We now show that calcein fluorescence reduces to a similar extent compared to our cytoplasmic GFP measurements during the phase-bright shrinkage measured before homeostatic extrusion and with hypertonic challenge (Fig. 1D). Both measurements are significantly with $P < 0.0001$.

4. Although they claim that apoptotic cell extrusions account for ~20-30% of extruding cells at steady state, the presentation of Supplemental Figure 2 is insufficient to support this. They can re-analyze the data to show how much percentage of cell extrusion is apoptotic.

This is a true statement but not what was measured in SFig. 2. We realize that this might have been confusing. SFig. 2B shows that HES does not occur prior to apoptotic cell extrusion, yet we do not measure the percent of apoptotic versus live cells extrusions in any of these graphs. Instead, we refer to our earlier work that documented the apoptotic extrusion rate in many cell types and in Fig. 1K, which shows that roughly 4 cells per 22 or roughly 20% total extrusions are apoptotic extrusion, matching previously reported rates.

5. Figure 4A indicates that the crowded monolayer shows a heterogeneous membrane polarity state. Is there any correlation between local cell density and membrane potentials at a single-cell level?

Generally, the plasma membrane potential decreases as cells crowd. However, as mentioned above, we decided to remove this content from this paper, as we found it distracts from our main point on the state of the single cells pre-extrusion. We are very interested in the membrane potential in background cells that are not extruding but feel it needs its own separate study, the more we investigated it.

6. In Supplementary Figure 4, Kv1.1 and Kv1.2 signals in separated and merged images look different, probably because they show other planes. The authors must show the same planes in each image. Also, they described that SWELL 1, Kv1.1, and Kv1.2 localize to the cell apex, but this is unclear from the represented images. To support this claim, they should show the images with higher magnification/resolution.

7. The concentration of amiloride should be described. It is not on the inhibitor table.

We have now added this. Thanks for noting.

- 1 Dwivedi, V. K. *et al.* Replication stress promotes cell elimination by extrusion. *Nature* **593**, 591-596 (2021). <https://doi.org:10.1038/s41586-021-03526-y>
- 2 Alberio, L. *et al.* A light-gated potassium channel for sustained neuronal inhibition. *Nat Methods* **15**, 969-976 (2018). <https://doi.org:10.1038/s41592-018-0186-9>
- 3 Atieh, Y., Wyatt, T., Zaske, A. M. & Eisenhoffer, G. T. Pulsatile contractions promote apoptotic cell extrusion in epithelial tissues. *Curr Biol* **31**, 1129-1140 e1124 (2021). <https://doi.org:10.1016/j.cub.2020.12.005>
- 4 Bortner, C. D. & Cidlowski, J. A. Ions, the Movement of Water and the Apoptotic Volume Decrease. *Front Cell Dev Biol* **8**, 611211 (2020). <https://doi.org:10.3389/fcell.2020.611211>

RESPONSE TO REVIEWERS

Referee #1 (Remarks to the Author):

In the revised manuscript, the authors have added a substantial body of new data to address the key question how cells are selected for extrusion. They now show that cell crowding causes Na⁺ entry via ENaC that causes pronounced depolarizations in those cells that – due to insufficient ATP production – are unable to extrude Na⁺ effectively via the Na⁺/K⁺ATPase. The resulting depolarization then activates the voltage-gated K⁺ and Cl⁻ channels KV1.1, KV1.2 and Swell1, respectively, causing cells to shrink. In cells with sufficient shrinkage, the downstream Piezo1-S1P-S1P2-Rho pathway becomes activated to initiate cell extrusion. The proposed mechanism suggests a “darwinistic” concept for the physiological maintenance of epithelial barrier integrity, in that the “weakest” cells become extruded, while simultaneously preventing an indiscriminate activation of the Piezo1-S1P-S1P2-Rho extrusion pathway in “stronger” epithelial cells.

Overall, the authors have done an excellent job to address my previous concerns. The manuscript has improved considerably and I only have a few minor concerns which you may please find below.

Thanks for your response, synopsis, and excellent feedback, which we feel improved the paper substantially.

Comments:

1. Can the authors ascertain that it is really depolarisation-dependent activation of Piezo1 that ultimately drives extrusion, or could it also be the loss of ATP per se? I realize that the authors show that inhibition of Piezo (and its downstream signalling) block extrusion, but this could also be an indirect effect vis Piezo-mediated ATP loss (by an unknown mechanism). This could be easily addressed with some control experiments showing that inhibition of Piezo does not prevent the loss of ATP and cell depolarisation during cell crowding.

This is an interesting idea. Following your suggestion, we have now immunostained and live imaged cells with ATP Red and DiBAC +/- Piezo1 inhibition by GsMTx4, which revealed that the early pathway ATP/depolarization steps are not disrupted by GsMTx4. ATP reduces, then depolarization and shrinkage occurs, but extrusion is blocked as expected, with Piezo1 inhibition. Only

cells that lose ATP depolarize and shrink. Since Piezo1 in all cells is inhibited with GsMTx4, yet only some cells lose ATP, Piezo1 does not mediate the ATP loss. Because Piezo1 is blocked, cells experiencing crowding cannot extrude, therefore, more depolarized cells accumulate over time (see A green v. black). Further, inhibition of Piezo1 with GsMTx4 did not block ATP reduction, indicating that Piezo1 is downstream both ATP loss and membrane depolarization (S. Fig. 5F&G). The text has been revised on line 244 to reflect this and we added the graph above to the supplemental figure: “Inhibition of Piezo1 with GsMTx4 did not block ATP reduction

or membrane depolarization indicating that Piezo1 is downstream in the extrusion pathway (S. Fig. 5F&G).

2. In figure 4G, it seems that mitochondrial shape (based on ATP red staining) changes substantially between homeostatic and crowded cells, which may reflect a response to the greater energy challenge in cell crowding. Is this a consistent finding? If so, it should probably be commented on.

Yes, we noted this, too, and will investigate this in future work. We do not yet know what it means and refrain from comment here, as we feel it might confuse the readers and dilute our message.

3. There is some inconsistency in the rebuttal letter as well as the manuscript with respect to the interpretation of the ENaC- and ATP-dependent changes in epithelial Na⁺ and the resulting membrane depolarisation. Specifically, at several instances in the rebuttal, the authors state that “decreasing ATP” or “blocking ATP synthase” causes “increased Na⁺ entry” – yet, according to the proposed concept, it is the lack of Na⁺ extrusion by the Na⁺/K⁺ATPase rather than an increased Na⁺ influx that causes the increase in cytosolic Na⁺ and the associated membrane depolarisation. Accordingly, in the revised manuscript in Fig. 4H, the y-axis should probably not read “average Na⁺ entry/1000 cells”, but rather “average increase in epithelial Na⁺ concentration/1000 cells”. Similarly, in line 246 the authors statement that “However, crowded cells challenged by ENaC activation with insufficient ATP levels to repolarize the membranes through the Na⁺/K⁺ ATPase will experience a current that activates KV1.1 and KV1.2, causing cell shrinkage via water egress” should probably be revised, as the activation of KV1.1 and KV1.2 is not the result of changes in Na⁺ current via ENaC, but due to changes in intracellular Na⁺ levels as a result of impaired Na⁺ extrusion.

Thank you for catching these inconsistencies, which could make the interpretation confusing. We have made the following changes:

- We believe that revising the y-axis was a good idea as it was confusing. Since the concentration here was not directly quantified, we relabel graphs 3H and 4G y-axes as: *Average Na⁺ positive cells per 1000 cells.*
- Line 238-40 is revised to state: *“Remarkably, ATP reduction on its own increased intracellular sodium, which crowding increased further (Fig. 4G&H), suggesting that crowding increases intracellular sodium, which cannot be pumped back out of cells with lower energy.*
- Line 255 was revised to state: *“However, crowding-dependent ENaC activation in cells with insufficient ATP needed to repolarize the membranes via the Na⁺/K⁺ ATPase will accumulate Na⁺ intracellularly, triggering KV1.1 and KV1.2 activation that causes cell shrinkage via water egress.”*

Referee #2 (Remarks to the Author):

The authors have properly revised the manuscript, adding the measurements of ATP and intracellular Na concentration. These data addressed how extruding cell are selected and support the “speculation” in the previous manuscript. Ex vivo experiments using mouse bronchial epithelial cells suggested the relevance of the identified mechanism in some physiological context. The title change seems reasonable to clearly show the main point of the study. However, I still have concern in the interpretation of the lightning assay, though overall responses to comments and discussion are reasonable.

Minor point

Authors still suggest the lightning assay can be a proxy for general volume loss. I agree, in most part, the lightning correlates to volume loss in their experiments (in the extrusion context they studied). However, it is shown in many papers apoptosis is accompanied with cell shrinkage in the epithelial monolayer, though authors describe “cell shrinkage is rare before apoptotic extrusion” based on the lightning assay. Hypertonic treatment failed to induce acute cell shrinkage when several channels were inhibited. The acute shrinkage can be observed even for a simple liposome. As authors could not show the detailed rationale of the lightning under cell shrinkage (only speculate), I don't think all cell shrinkage should be detected by the lightning assay. I think it is better to carefully suggest the applicable subject for the lightning assay, otherwise authors should further investigate the generality of a one-to-one correspondence between the lightning and cell shrinkage in other contexts.

Thanks very much for this comment and your previous review, which we feel greatly improved the paper. We agree that a ‘lightning’ response does not always represent cell shrinkage and could reflect different states. In this paper, however, we use it as a proxy for consistent changes in volume in our assay to enable more robust quantification. As water regulation is becoming an important regulator of many processes, we hope that this lightning will trigger other investigators to also examine water regulation, by following up with the assays we mention. We tried to reflect this by adding the sentence below on line 279.

We have made the following edits to reflect this:

Line 163 : “using our lightning assay and calcein to measure cell shrinkage in our assay in...”

In Line 274 of the discussion, we add: “*In this study, application of current also caused a lightening of the junctions, yet we note that this response can result from other perturbations causing enhanced light between cell-cell junctions and does not always necessarily reflect cell shrinkage.*”

Referee #3 (Remarks to the Author):

Overall, the modifications made by the authors nicely improve the manuscript and the impossibility to perform some of the experiments asked well justified (such as aquaporin inhibition for example).

Many thanks for your previous input, which we believe greatly improved our study. We respond below to your outstanding questions.

I still have a few minor comments that should be easy to address:

- In figure 4I: Osmolarity seems weirdly represented in the shrinking cell. I would have expected more electrolytes in the environment than in the extruding cell to explain the loss of water.

Thank you for pointing this out. We have revised this to show that although K^+ and Cl^- egress with water that Na^+ increases in the cell that will extrude. The model has been revised to include a higher number of Na^+ ions in the depiction.

-Extrusions are not easy to spot in bronchiolar epithelial cells of mouse lung (Fig4E-F). What is the readout of extrusion here? How is extrusion quantified? The images could be improved to be more convincing.

Thank you for the feedback. The lung slices are roughly 200 micron thick, making them difficult to image by phase at low magnification, needed for our timelapse analysis with ATP red and DiBAC. While they are easier to see as stills in fixed sections, included in Figure 1M, where actin-myosin rings are at the basolateral surface of the cell being extruded, we have developed methods to analyse these by phase for this and our asthma paper (Bagley et al, Science 2024). We describe this in the methods section entitled Extrusion, by ensuring that cells that squeeze out of the tissue reintegrate back into the tissue, if played in reverse.

Line 706: *'These cells were then followed backward in time to quantify cell shrink, compared to initiation of extrusion. By contrast, cells that round up, divide and reincorporate into the monolayer were scored as mitoses. The cells that were already eliminated by extrusion at the beginning of filming were excluded from our quantifications.'*

- Fig3D: Na^+ entry in future extruding cells should be quantified together with cell shrinkage to better highlight the sequence of events (Na^+ entry first, shrinkage next). A representative example is insufficient.

Thanks, and importantly this suggestion gave us the idea to add the following graph. We have added a quantification of when Na⁺ entry occurs, using CoroNa AM (the sodium dye), with respect to the homeostatic early shrink (HES) (S. Figure 5F and below). This graph indicates that Na⁺ enters on average 11 minutes before shrinkage. On line 241-244: “Although depolarization and sodium entry could not be filmed together, sodium entry typically occurs before depolarization (when comparing S. Fig. 5G with Fig. 4B times before shrinkage), suggesting that Na⁺ intracellular accumulation leads to depolarization.”

- When talking about the impact of cell contractility on shrinkage, the author mention: Thus, contractility appears to suppress cell shrinkage, presumably by stabilizing cell-cell junctions.” Since there could be many other ways for contractility to prevent shrinkage, the last part of the sentence should be removed.

Good point. We have removed this.

Referee #4 (Remarks to the Author):

The authors have revised the manuscript in accordance with the suggestions provided by the various referees. They now present additional evidence indicating that the energy/ATP level of individual cells influences their likelihood of being extruded. While this represents an interesting extension to the proposed model of cell extrusion, it also raises several major concerns: Thanks for your input that helped greatly improve our manuscript.

- The finding that uniformly lowering ATP levels results in increased cell extrusion is perhaps not entirely unexpected. A more compelling approach would be to manipulate ATP levels in a mosaic fashion and to test whether uniformly increasing ATP levels correspondingly reduces the rate of cell extrusion.

During normal homeostasis, as you will note, the ATP levels within cells are naturally lowered mosaically throughout the monolayer. We like the idea, however, of increasing overall ATP levels, which we have done by adding fresh medium with extra glucose where we see ATP levels rapidly increase quantified below, which drastically reduces the rates of extrusion (graph on left, added to S. Figure 5E. Text was also revised on line 235 to include: “Conversely, supplementing cells with

additional glucose increased ATP levels and decreased the rate of extrusion (S. Fig. 5E-F)."

- The authors suggest—but do not directly demonstrate—that ATP is required for Na⁺/K⁺-ATPase activity to normalize membrane potential. How does reduced Na⁺/K⁺-ATPase activity affect cell extrusion? How sensitive is this enzyme's activity to fluctuations in intracellular ATP levels? Furthermore, could any potential reduction in cell extrusion due to elevated ATP levels be reversed by inhibiting Na⁺/K⁺-ATPase?

We mentioned in our response to reviewers that we tried using ouabain to reduce NA⁺/K⁺ ATPase, however, this immediately killed all the cells in the monolayer, further supporting its importance, but making it difficult to manipulate its levels. We have added "Inhibition of Na⁺/K⁺ pump with ouabain, however, rapidly killed all epithelial cells, indicating is essential function (data not shown)." to the main text starting on line 237.

- Is cell extrusion governed by the absolute ATP level within a cell, or its relative level compared to neighbouring cells?

While we are not aware if ATP Red and Queen 37 can measure absolute ATP levels, we noted in movies that it is more likely to be relative to neighbours, as seen with the depolarization, measured with DiBAC. In figure 4 B, we show that cells do not extrude do not change relative ATP levels over time.

- What is the relationship between the degree of Na⁺ influx/ membrane depolarization and intracellular ATP levels?

This is a very interesting point. CoroNa Am which is the intercellular Na⁺ dye does not give absolute values, but we do note that cells with relatively fluorescence more than their neighbours extrude, much like the relative levels noted with DiBAC increase and ATP Red and Queen37 decreases, suggesting that cells are measuring relative levels compared to neighbouring cells. We do not include this as a statement, as we are not sure we can back it up absolutely.

Given ATP's general role in tissue homeostasis and cell survival, its specific contribution to crowding-induced cell extrusion needs to be more clearly demonstrated.

We agree that the role of ATP is imperative to tissue homeostasis and cell survival. This work has inspired many new directions and first author, Saranne Mitchell, plans to further investigate these important roles of energy and tissue homeostasis in future work. We believe that the new included data better supports the important role of energy deficiency in triggering crowding-induced extrusion, which will be important for future studies.

This email has been sent through the Springer Nature Manuscript Tracking System NY-610A-SN&MTS

Confidentiality Statement:

This e-mail is confidential and subject to copyright. Any unauthorised use or disclosure of its contents is prohibited. If you have received this email in error please notify our Manuscript Tracking System Helpdesk team at <http://platformsupport.nature.com> .

Details of the confidentiality and pre-publicity policy may be found here <http://www.nature.com/authors/policies/confidentiality.html>

Privacy Policy | Update Profile